# The IRE1β-mediated unfolded protein response is repressed by the chaperone AGR2 in mucin producing cells

Lisa Neidhardt [1✉], Eva Cloots [2,3], Natalie Friemel [1], Caroline A M Weiss [1], Heather P Harding [1], Stephen H McLaughlin [4], Sophie Janssens [2,3] & David Ron [1✉]

## Abstract

**Effector mechanisms of the unfolded protein response (UPR) in the endoplasmic reticulum (ER) are well-characterised, but how ER proteostasis is sensed is less well understood. Here, we exploited the beta isoform of the UPR transducer IRE1, that is specific to mucin-producing cells in order to gauge the relative regulatory roles of activating ligands and repressing chaperones of the specialised ER of goblet cells. Replacement of the stress-sensing luminal domain of endogenous IRE1α in CHO cells (normally expressing neither mucin nor IRE1β) with the luminal domain of IRE1β deregulated basal IRE1 activity. The mucin-specific chaperone AGR2 repressed IRE1 activity in cells expressing the domain-swapped IRE1β/α chimera, but had no effect on IRE1α. Introduction of the goblet cell-specific client MUC2 reversed AGR2-mediated repression of the IRE1β/α chimera. In vitro, AGR2 actively de-stabilised the IRE1β luminal domain dimer and formed a reversible complex with the inactive monomer. These features of the IRE1β-AGR2 couple suggest that active repression of IRE1β by a specialised mucin chaperone subordinates IRE1 activity to a proteostatic challenge unique to goblet cells, a challenge that is otherwise poorly recognised by the pervasive UPR transducers.**

**Keywords** Endoplasmic Reticulum (ER); Molecular Chaperones; Mucin; Protein Multimerisation; Unfolded Protein Response (UPR)
**Subject Categories** Digestive System; Post-translational Modifications & Proteolysis; Translation & Protein Quality

See also: E. Cloots et al (2023)

## Introduction

In eukaryotes, the endoplasmic reticulum (ER) traffics most proteins destined for secretion or membrane insertion (Uhlén et al, 2015). The ER is equipped with a specialised protein folding and processing machinery constituting the folding capacity of the compartment.

Capacity is matched to the inward flux of newly synthesised proteins by signalling pathways, jointly referred to as the Unfolded Protein Response (UPR). UPR stress-sensors monitor the balance between folding load and folding capacity in the ER and their effectors restore proteostasis to counteract ER stress. The mammalian UPR has three known signalling branches, each headed by a unique signal-transducing ER-resident transmembrane protein: IRE1, PERK and ATF6 [reviewed in Walter and Ron, 2011].

The most conserved of these signal transducers is Inositol Requiring Enzyme 1 (IRE1), a type I transmembrane protein that monitors protein-folding load with its stress-sensing luminal domain (LD) (Cox et al, 1993; Mori et al, 1993). Luminal ER stress is communicated across the ER membrane to favour dimerisation/oligomerisation-dependent autophosphorylation of IRE1's cytosolic kinase–endonuclease extension (KEN) effector domain. Downstream signalling adjusts folding capacity through chaperone expression, ER membrane biogenesis and regulation of protein flux [reviewed in Walter and Ron, 2011]. Two biochemical processes contribute to this outcome: IRE1's cytosolic RNase domain initiates splicing of the mRNA encoding the transcription factor X- box binding protein (XBP)-1 (or HAC1 in yeast) (Cox and Walter, 1996; Calfon et al, 2002; Yoshida et al, 2001). Active IRE1 also cleaves other mRNAs in proximity to the ER membrane in a process termed Regulated IRE1-dependent Decay (RIDD) (Hollien and Weissman, 2006; Hollien et al, 2009). Whereas downstream IRE1 signalling is relatively well-characterised, the molecular details of the luminal stress-sensing mechanism leading to IRE1 activation remain incompletely understood.

Two models describe how IRE1 LD senses proteostasis: A direct binding model proposes unfolded proteins as activating ligands that stabilise IRE1's dimeric/oligomeric active state. This model draws on structural similarity between the yeast (y) IRE1 LD and the major histocompatibility peptide-binding complexes (MHCs) suggesting a peptide-binding groove that spans the yIRE1 LD dimer interface (Credle et al, 2005). A similar groove is present in the alpha isoform of human IRE1 (hIRE1α LD) however, it is too narrow to accommodate a peptide (Zhou et al, 2006). Whilst endogenous IRE1 ligands have yet to be identified, experimentally

[1]Cambridge Institute for Medical Research, University of Cambridge, Cambridge CB2 0XY, UK. [2]Laboratory for ER stress and Inflammation, VIB Center for Inflammation Research, Technologiepark-Zwijnaarde 71, 9052 Ghent, Belgium. [3]Department of Pediatrics and Internal Medicine, Faculty of Medicine and Health Sciences, Ghent University, Technologiepark-Zwijnaarde 71, 9052 Ghent, Belgium. [4]MRC Laboratory of Molecular Biology, Francis Crick Avenue, Cambridge CB2 0QH, UK. ✉E-mail: ln327@cam.ac.uk; dr360@cam.ac.uk

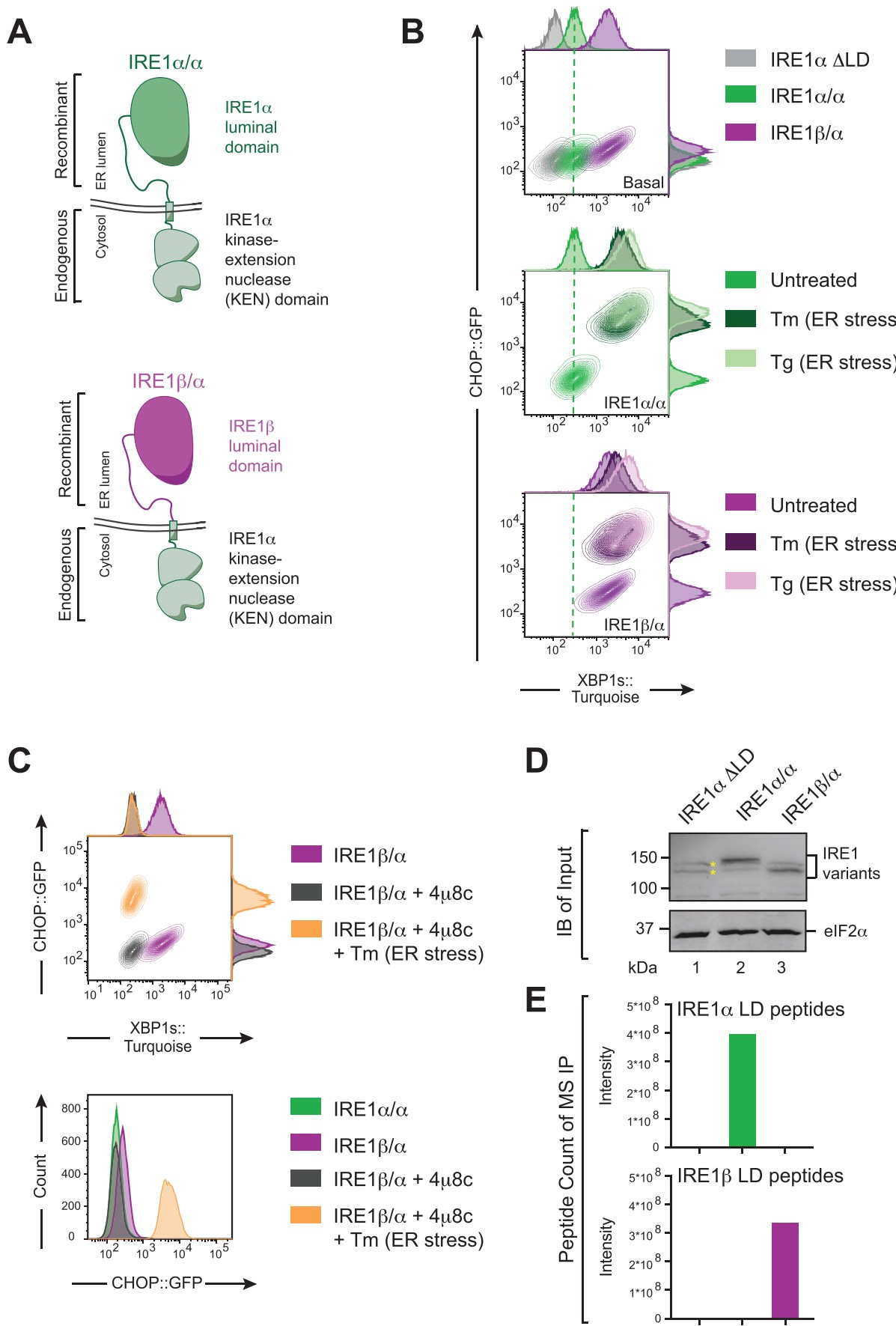

**Figure 1. An IRE1 luminal domain (LD) swap to study IRE1β in the heterologous system of Chinese Hamster Ovarian (CHO) cells.**

(A) Schematic depiction of the chimeric proteins consisting of IRE1α or β LD, endogenous IRE1α transmembrane and cytosolic domains that were knocked into the endogenous *ERN1* locus of dual UPR reporter CHO cells via CRISPR-Cas9 technology. (B) Two-dimensional contour plots of CHOP::GFP and XBP1s::Turquoise signals from dual UPR reporter CHO cells stably expressing the indicated IRE1 variants from the endogenous *ERN1* locus untreated and treated with the ER stressors tunicamycin (Tm) or thapsigargin (Tg). Clones used for the analysis were derived from an IRE1α LD null parental cell line [IRE1α ΔLD, previously described in Kono et al, 2017]. A representative data set out of three independent experiments is shown. (C) Contour plot as in (B, top panel) and histogram of CHOP::GFP intensity (bottom panel) from dual UPR reporter CHO cells with the indicated genotype treated with the IRE1 inhibitor 4μ8c (Cross et al, 2012) and Tm. A representative data set out of three independent experiments is shown. (D) Representative anti-IRE1α cytosolic domain immunoblot (IB) of whole cell lysates from cells described in (B) expressing the indicated IRE1 variants. The signal arising from the IRE1α/α and IRE1β/α chimeras is indicated as is the background signal (yellow asterisks). (E) Bar diagram of the number of peptides mapping to either the IRE1α or IRE1β LD counted by mass spectrometry (MS) analysis of tryptic digests of material immunoprecipitated (IP) with anti-IRE1α KEN-domain antibodies from samples shown in (D). Source data are available online for this figure.

determined peptides can induce a shift towards higher-order active species of both the yeast and human IRE1α LD in vitro (Gardner and Walter, 2011; Karagöz et al, 2017). However, caveats apply to the strength of the data linking these biochemical features to peptide binding in the MHC-like groove (see Amin-Wetzel et al, 2019; Fig. 2—Figure Supplement 2).

An alternative model proposes that IRE1 is repressed by the ER-localised heat-shock protein (Hsp70) chaperone, BiP. BiP binding to IRE1's stress-sensing LD disfavours an active oligomeric state. Unfolded proteins compete for BiP, thereby kinetically disrupting the inhibitory IRE1-BiP complex. This chaperone inhibition model fits the inverse correlation between stress-induced IRE1α activity and the recovery of BiP in complex with it (Bertolotti et al, 2000; Okamura et al, 2000; Oikawa et al, 2009). It is also supported by evidence that BiP's interaction with the IRE1α LD disfavours the latter's active oligomeric state in vitro (Amin-Wetzel et al, 2017) and by the observation that targeting BiP to IRE1α LD in cells inhibits the latter's activity (Amin-Wetzel et al, 2019). However, establishing the relevance of BiP repression in cells is impeded by the difficulty in deconvoluting BiP's direct action from its indirect effects on IRE1 activity, arising from BiP's role in maintaining ER proteostasis.

Vertebrates express two IRE1 isoforms [reviewed in (Cloots et al, 2021)]. The well-studied IRE1α isoform (encoded by the *ERN1* gene) is ubiquitously expressed and serves as a UPR stress sensor in most cells (Tirasophon et al, 1998). IRE1β [encoded by *ERN2* (Wang et al, 1998)] expression is restricted to the epithelium of the gastrointestinal tract and airways (Bertolotti et al, 2001; Tsuru et al, 2013). Single-cell analysis showed that IRE1β transcripts are particularly enriched in goblet cells that produce protective mucin (MUC) glycoproteins (Haber et al, 2017).

IRE1β is involved in mucin production (Martino et al, 2013) since mice lacking IRE1β are hypersensitive to challenges that require intact goblet cell function (Bertolotti et al, 2001), resulting in phenotypic overlap with mice lacking MUC2 (the prevailing mucin in colonic epithelium) (Van der Sluis et al, 2006). Accumulation of aberrant MUC2 and mucin dysgenesis in IRE1β deficient mice suggests an unmet proteostatic challenge (Heazlewood et al, 2008; Tsuru et al, 2013). This notion is further supported by the phenotypic overlap of the IRE1β deficient mice with mice lacking AGR2, a key mucin chaperone (Park et al, 2009; Zhao et al, 2010). IRE1β signalling thus appears to be tailored to the specialised requirements of ER proteostasis in goblet cells.

Goblet cells are mucin-producing factories (Birchenough et al, 2015). Mucin biogenesis entails the formation of many intra- and interchain disulphide bonds concentrated in the cysteine-rich amino (N)-terminal and carboxyl (C)-terminal domains (Perez-Vilar and Hill, 1999; Godl et al, 2002; Lidell et al, 2003), a process that requires AGR2, the aforementioned specialised small protein disulphide isomerase (PDI) family member. PDIs are thiol-sensitive folding catalysts of the ER and have been implicated in ER stress regulation. Previously, PDIA6 was reported to control the decay of IRE1α signalling as ER stress wanes (Eletto et al, 2014, 2016), and PDIA1 phosphorylation has been implicated in attenuating IRE1α signalling (Yu et al, 2020).

If goblet cell-specific proteins like AGR2 or MUC2 are regulators of the IRE1β isoform, a domain swap of the endogenous IRE1α with IRE1β LD (in a heterologous *ERN1*-dependent cell culture system lacking components present in goblet cells) might serve as a platform to study both activating ligands and repressive agents by complementation. Furthermore, swaps in the endogenous *ERN1* locus might also shed light on specialisation of IRE1β's cytosolic KEN effector domain. Previously, some report on near parity between the two isoforms in XBP1 splicing (Feldman et al, 2019), whilst others report IRE1β to have a weaker splicing activity and enhanced RIDD activity (Imagawa et al, 2008). This study, and an accompanying manuscript (Cloots et al, 2023), report on the adoption of Chinese Hamster Ovary (CHO) cells with dual UPR reporters of the IRE1 and PERK branches to study IRE1β specialisation and thereby illuminate fundamentals of UPR stress sensing and signalling.

## Results

### Divergent-regulatory machineries for IRE1α and β

To compare the IRE1α and IRE1β stress-sensing luminal domains (LDs), we turned to a previously established CHO cell line stably expressing XBP1s::Turquoise and CHOP::GFP reporters (controlled exclusively by the IRE1 and mostly by the PERK UPR branches, respectively). A genomic deletion encompassing the IRE1α LD-encoding exons 2–12 was introduced to inactivate the endogenous *ERN1*. This IRE1α ΔLD cell line lost all XBP1 splicing, consistent with the *ERN1*-dependence of XBP1 splicing and the inactivity of the endogenous IRE1β encoding *ERN2* in CHO cells (Kono et al, 2017). Using CRISPR-Cas9 directed homologous recombination, we reconstituted the endogenous deleted locus with either the IRE1α or IRE1β murine LD (Fig. 1A; Appendix Fig. S1A). Flow cytometry showed that reconstitution of the deleted locus

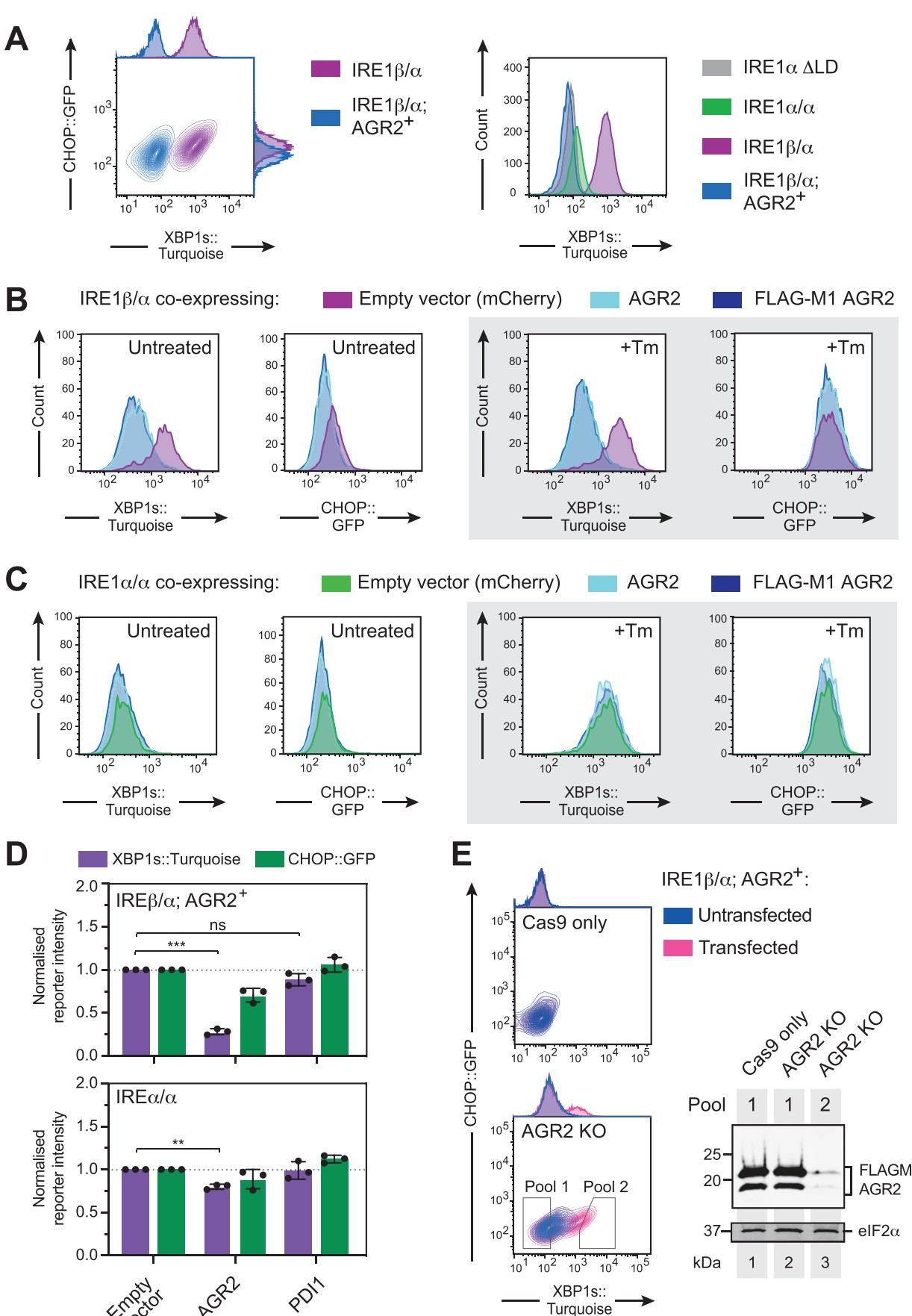

**Figure 2. AGR2 selectively represses the IRE1β luminal domain (LD).**

(A) Left panel: Two-dimensional contour plot of CHOP::GFP and XBP1s::Turquoise signals from dual UPR reporter CHO cells stably expressing the IRE1β/α protein with and without stably co-expressed AGR2. Right panel: Histogram of XBP1s::Turquoise intensity of dual UPR reporter cells with the indicated genotype. A representative data set out of three independent experiments is shown. (B) Histograms of CHOP::GFP and XBP1s::Turquoise intensities from dual UPR reporter cells stably expressing IRE1β/α transiently transfected with mCherry-marked expression plasmids encoding untagged and FLAG-M1-tagged AGR2. The grey box marks data obtained after ER stress induction by tunicamycin (Tm). A representative data set out of three independent experiments is shown. (C) As in (B) but from cells expressing IRE1α/α. (D) Quantification of reporter signals from IRE1β/α and IRE1α/α expressing dual UPR reporter CHO cells transiently transfected with AGR2 or PDI1 (mCherry as expression marker). Shown are the mean ± SD of three independent repetitions. Statistical analysis was performed by two-sided unpaired Welch's *t* test and significance is indicated by asterisks (ns *P* > 0.1, **P* < 0.01, ***P* < 0.001). (E) Contour plots as in (A) of IRE1β/α and AGR2 expressing cells (IRE1β/α; AGR2[+]) transfected with a guide targeting the AGR2 transgene. XBP1s::Turquoise low (pool 1) and high (pool 2) populations were sorted and whole cell lysates analysed for AGR2 (FLAG-M1) on western blot. Source data are available online for this figure.

with IRE1α LD restores both basal levels of the XBP1s::Turquoise reporter and its responsiveness to ER stress (induced by the N-glycosylation inhibitor, tunicamycin or the calcium pump inhibitor, thapsigargin). Reconstituting the deleted locus with the IRE1β LD had a different profile: with high basal activity of the XBP1::Turquoise reporter and modest further inducibility upon ER stress (Fig. 1B).

Reporter activation was dependent on IRE1's RNase activity, as it was blocked by the IRE1 inhibitor 4μ8C (Cross et al, 2012). This applied not only to the XBP1::Turquoise reporter but also to the slightly enhanced basal activity of the CHOP::GFP reporter observed in the IRE1β LD reconstituted cells, consistent with the known contribution of IRE1 to CHOP expression (Wang et al, 1998) (Fig. 1C). The experimental system was further validated by the observation that the XBP1s::Turquoise reporter mirrored endogenous XBP1 mRNA splicing (Appendix Fig. S1B). This leads to the conclusion that the chimeric IRE1β/α protein, which is expressed at levels similar to IRE1α/α (Fig. 1D,E), is basally de-repressed.

CHO cells likely lack goblet cell-specific IRE1β LD-activating ligands. Therefore, the constitutive activity of IRE1β/α is most readily explained by lack of an IRE1β LD specific repressor. AGR2, a mucin-selective ER-localised chaperone (Haber et al, 2017), was proposed as a candidate for the missing ingredient based on two observations: (1) IRE1β is co-expressed with AGR2 in goblet cells, (2) AGR2 over-expression was noted to blunt the toxic consequences of IRE1β over-expression in cultured cells (see accompanying manuscript Cloots et al, 2023).

To test AGR2's ability to regulate IRE1β LD, we introduced it by retroviral transduction. AGR2 strongly suppressed IRE1β/α activity indicated by reduced XBP1s::Turquoise levels (Fig. 2A). Suppression was observed both by unmodified AGR2 (expressed with its endogenous signal peptide) and with FLAG-M1-tagged AGR2, targeted to the ER with a heterologous signal peptide (Fig. 2B; the latter, which is reactive with anti-FLAG-M1 antibodies, was used in subsequent experiments). In contrast, AGR2 expression failed to suppress the stress-induced expression of the PERK-dependent CHOP::GFP reporter in the same cells (Fig. 2B) and had only minimal effect on signalling in IRE1α/α cells (Fig. 2C,D). Expression of PDIA1, a different member of the PDI family, failed to repress IRE1β/α signalling (Fig. 2D). Together, these observations attest to the selectivity of AGR2's repressive effect on IRE1β LD. Importantly, elevated basal XBP1 splicing activity of the IRE1β/α chimera was restored upon CRISPR-Cas9-mediated knockout of the AGR2

encoding sequence, confirming the dependence of repression on the expression of the AGR2 protein in the IRE1β/α; AGR2[+] cells (Fig. 2E).

Taken together, these observations suggest that IRE1β LD has evolved with its own set of selective repressors, one of which is AGR2. To explore this further, we characterised the putative interaction partners in vitro.

## AGR2 binds IRE1β's luminal domain

The IRE1β LD-AGR2 interaction was reconstituted in vitro using recombinant proteins purified from bacteria and Bio-Layer Interferometry (BLI) to measure binding. IRE1β LD was immobilised on the BLI probe and sequentially exposed to increasing concentrations of AGR2. This resulted in reversible association and dissociation traces (Fig. 3A). Plotting the BLI signal against AGR2 concentration revealed a $K_{1/2\ max}$ of binding of 19 μM (Fig. 3B). In contrast, immobilised IRE1α LD generated a weak binding signal with AGR2, consistent with a minimal repressive effect of AGR2 in IRE1α/α cells (Fig. 3A).

The BLI assays were conducted under reducing conditions that preclude a role for thiol exchange in the IRE1β LD-AGR2 interaction measured. However, since AGR2 is a member of the PDI family, we wondered whether AGR2's repressive effect on the IRE1β/α chimera involves the formation of a mixed disulphide with IRE1β LD as its client. Returning to the homologous recombination platform in the CHO UPR reporter cells, we reconstituted the deleted endogenous *ERN1* locus with an IRE1β LD lacking its cysteines (ΔC). This resulted in lower basal XBP1 splicing signal than observed in wild-type IRE1β LD reconstituted cells (Fig. EV1A), but the IRE1β ΔC/α chimera remained responsive to AGR2 repression, albeit less so than the wild-type IRE1β/α chimera (Fig. EV1B). BLI measurements confirmed an interaction between IRE1β LD ΔC and AGR2, however with reduced binding affinity (Fig. EV1C).

To further explore the possibility of mixed disulphide formation between IRE1β LD and AGR2, cell lysates were treated with N-ethylmaleimide (NEM) that blocks free thiols and preserves mixed disulphides. Immunoprecipitates (IP) of IRE1β/α (expressed from the endogenous *ERN1* locus) had no AGR2 signal. Therefore, we turned to over-expression. In this experimentally-sensitised system co-expression of full-length IRE1β and FLAG-M1-tagged AGR2 yielded robust co-IP. Yet, no mixed disulphide complexes were detected, neither in samples from whole cell extracts nor upon enrichment after IP. Instead, IRE1β co-immunoprecipitated

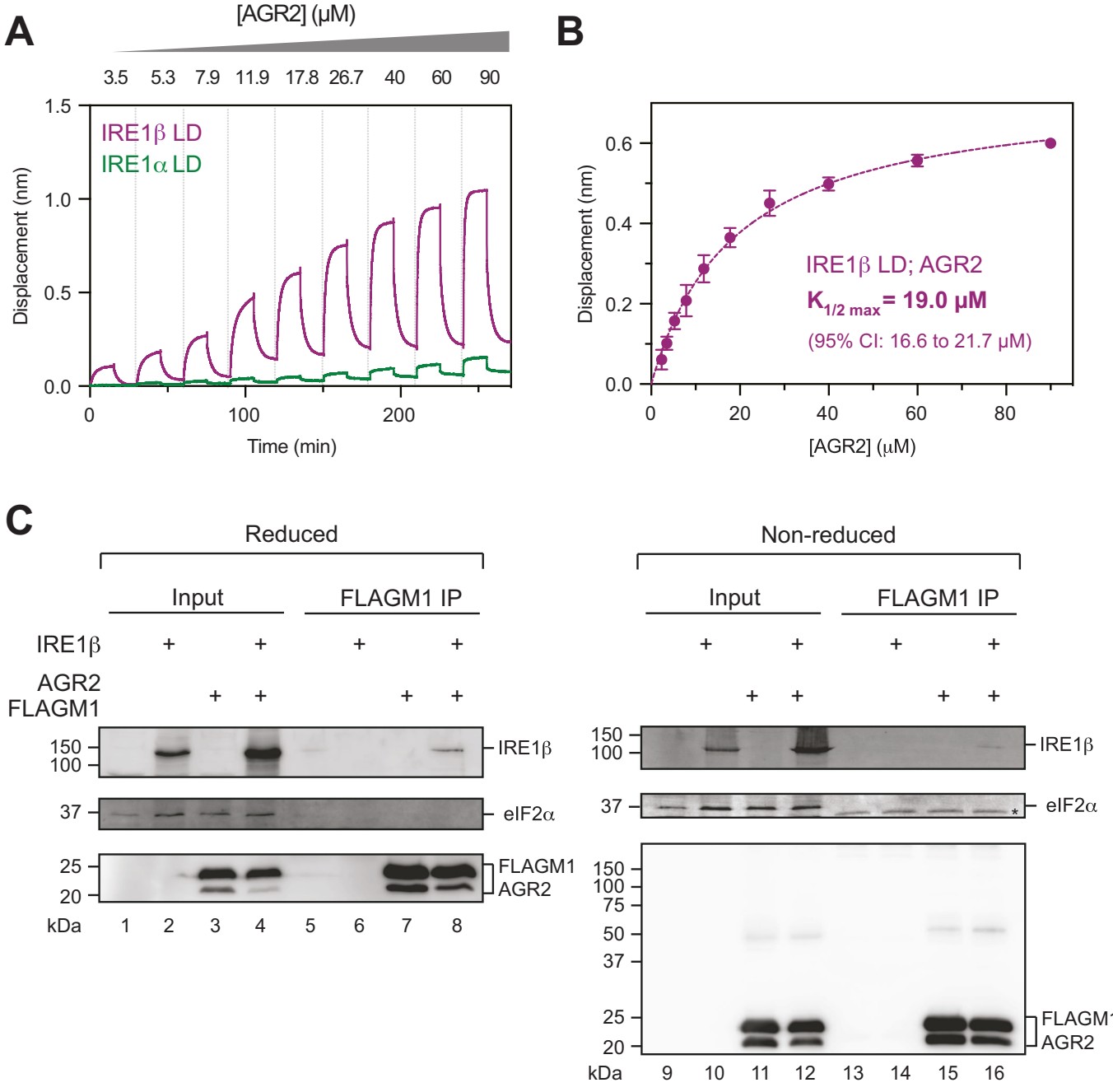

**Figure 3. AGR2 selectively binds IRE1β's luminal domain (LD).**

(A) Bio-Layer Interferometry (BLI)-derived association and dissociation traces of streptavidin sensors loaded with the indicated biotinylated ligands and exposed sequentially to increasing concentrations of AGR2. A representative experiment of three independent repetitions is shown. (B) BLI signal from an IRE1β LD probe as a function of the concentration of interacting AGR2 fitted to a one-site-specific binding function with the indicated confidence interval (CI). Shown are the mean ± SD of three independent repetitions. (C) Representative immunoblots of whole cell lysates prepared in presence of N-ethylmaleimide (NEM) from wild-type CHO cells overexpressing full-length IRE1β and FLAGM1 AGR2. Lysates served as an input for immunoprecipitation reactions (IP) loaded onto a reducing ('reduced', left panel) or non-reducing ('non-reduced', right panel) SDS-PAGE. The asterisk (*) marks a non-specific band. Source data are available online for this figure.

with FLAG-M1 AGR2 in the presence of reducing agents (Fig. 3C). In summary, both the in vitro measurements (conducted in the presence of reductants) and the observations made in vivo reported on a direct interaction between IRE1β LD and AGR2 that occurred independently of a mixed disulphide.

## AGR2 binding favours IRE1β luminal domain monomers

Given the importance of oligomerisation to IRE1 activity (Shamu and Walter, 1996), we tested whether IRE1β LD's dimeric/oligomeric state was altered by interaction with its repressor

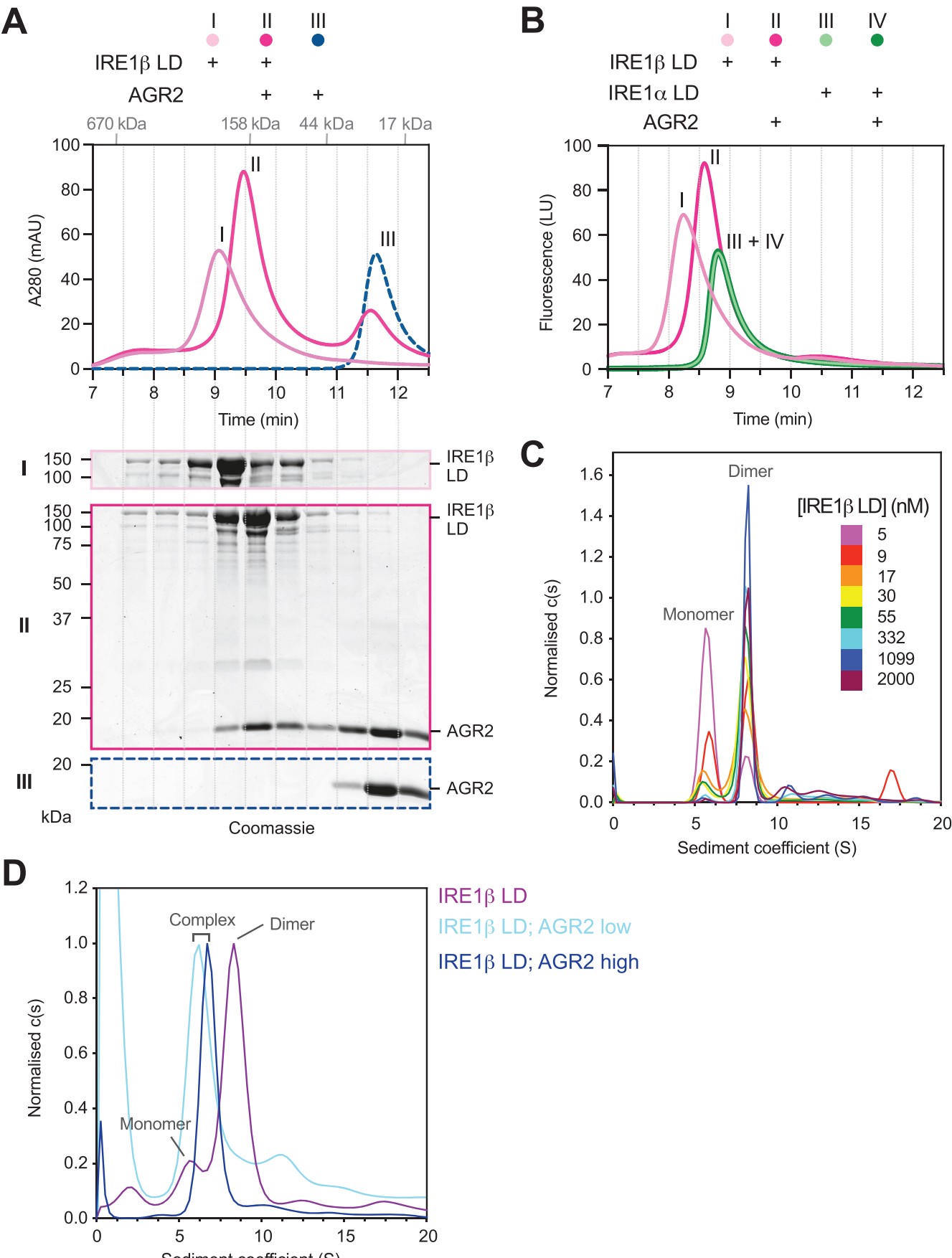

**Figure 4.  AGR2 favours an IRE1β luminal domain (LD) monomer.**

(A) Upper panel: Size-exclusion chromatography (SEC) elution profiles [protein absorbance at 280 nm (A280)] plotted against elution time of GFP-tagged IRE1β LD with and without AGR2. Elution times obtained from reference marker proteins are noted in grey. Every 30 s, SEC fractions were collected and analysed for their protein content via SDS-PAGE. Lower panels: Coomassie-stained SDS-PAGE of fractions from each chromatogram. Representative data of three independent experiments is shown. (B) SEC elution profiles as in (A), comparing fluorescently labelled IRE1β and IRE1α LD in the presence and absence of AGR2. Representative plot of three independent experiments is shown. (C) Analytical ultracentrifugation analysis of GFP-tagged IRE1β LD in solution. The c(s) distribution plots of increasing concentrations of IRE1β LD are shown. (D) As (C) but showing plots of GFP-tagged IRE1β LD at a concentration of 11.2 nM in the absence and presence of 1 μM (low) and 30 μM (high) AGR2. Source data are available online for this figure.

AGR2 in vitro. Size-exclusion chromatography (SEC) of escalating concentrations of IRE1β LD suggested that like the alpha isoform, it too exists in a dynamic equilibrium between monomers and dimers/oligomers, as a concentration-dependent shift of the peak elution time was observed (Fig. EV2A,B). The smooth transition to later elution times upon dilution and lack of distinct monomer and dimer/oligomer peaks suggested an exchange between species in the timescale of a SEC experiment (i.e., minutes). An orthogonal method, mass photometry by interferometric scattering microscopy (ISCAT), showed that IRE1β LD species in solution correspond in mass to that expected of monomers and dimers (Fig. EV2C).

Incubation of IRE1β LD with AGR2 prior to SEC analysis resulted in a delay of the peak elution, suggesting an increased fraction of monomers. This was observed both by monitoring protein absorbance at 280 nm (Fig. 4A) and by selectively tracking the elution of fluorescently tagged IRE1β LD by its fluorescence (Fig. 4B). Importantly, the shift of IRE1β LD to later eluting fractions upon exposure to AGR2 coincided with an opposite shift of AGR2 to earlier-eluting fractions, consistent with a complex between monomeric IRE1β LD and AGR2 (Fig. 4A, lower panel). A similar result was observed when AGR2 was incubated with the cysteine-free IRE1β LD ΔC, in line with the cell-based and BLI data suggesting that a mixed disulphide is dispensable for the repressive interaction (Fig. EV2D). Fluorescently tagged IRE1α served as a negative control and confirmed the absence of a monomerising effect of AGR2 (Fig. 4B).

As an orthogonal approach, we performed sedimentation velocity analytical ultracentrifugation (SV-AUC). Two main species were observed at 8.2 S, corresponding to the IRE1β LD dimer ($S_{w,20} = 9.2$ S with a calculated mass of 234 kDa and a frictional coefficient of 1.459) and 5.7 S, corresponding to the monomer ($S_{w,20} = 6.4$ S with a calculated mass of 116 kDa and a frictional coefficient of 1.318) (Fig. 4C). As observed in SEC, the concentration-dependent change in the monomer/dimer ratio was consistent with a monomer–dimer equilibrium and a $K_D$ of dimerisation in the nanomolar range. The apparent discrepancy between the $K_{1/2\ max}$ of the monomer to dimer transition assessed by SEC (Fig. EV2B) and AUC might be explained by the dissociation of a dynamic complex in the former.

Incubation of IRE1β LD with AGR2 prior to SV-AUC resulted in only a single species at 6.6 S ($S_{w,20} = 7.5$ S) at 1 μM and 6.9 S ($S_{w,20} = 7.8$ S) at 30 μM (Fig. 4D). These intermediate sedimentation velocities fell between the monomeric and dimeric species and had calculated masses of 140 and 149 kDa (for 1 μM and 30 μM AGR2 samples, respectively) suggesting that one or two AGR2 molecules were bound per IRE1β LD monomer. Consistent observations were made by ISCAT: When incubated with AGR2, a third peak appeared (in addition to the monomer and dimer peak of wild-type or IRE1β LD ΔC), consistent in size with monomeric

IRE1β LD bound by one or two molecules of AGR2 (the resolution of the measurement is inadequate to distinguish between the two, Fig. EV2E).

## IRE1β repression is affected by the state of AGR2's active site and involves destabilisation of luminal domain dimers

Despite the lack of a resolving cysteine (a feature shared with other PDI's, such as ERp44 and TMX5) AGR2 has a well-formed thioredoxin domain (Persson et al, 2005). Given the evidence that the thioredoxin-like active sites of PDIs can allosterically modulate chaperone activity (Serve et al, 2010; Wang et al, 2012; Okumura et al, 2019) we compared repression of IRE1β by wild-type AGR2 with that of active site C81 mutants. Triple-channel flow cytometry following intracellular staining of FLAG-M1-tagged AGR2 correlated its abundance with IRE1β/α activity in transiently transfected CHO cells. At comparable levels of expression, the C81S and C81E active site mutants repressed IRE1β/α-mediated XBP1 splicing less effectively than the wild-type. A C81W mutant was completely inactive as a repressor and even stimulated reporter activity at the highest levels of expression. The latter likely reflects a non-specific challenge to ER proteostasis, as it was also observed in the CHOP::GFP channel (Fig. 5A,B).

At the concentrations normally present in the ER of goblet cells (~460 μM, see below), AGR2 is likely a dimer [given a dimerisation $K_D$ of 8 μM (Patel et al, 2013)]. To test the effect of AGR2's dimeric state on its ability to repress the chimeric IRE1β/α, we expressed a monomeric E60A mutant (Patel et al, 2013). It too was compromised in repressing IRE1β/α activity (Fig. 5A,B).

Impaired repression of IRE1β/α in transfected cells correlated with weaker interaction of purified C81S and C81E AGR2 mutants with the IRE1β LD in the BLI assay (Fig. 5C). The purified AGR2 C81W and E60A mutants bound irreversibly to the IRE1β LD BLI probe yielding uninterpretable binding curves.

Impaired binding of AGR2 C81S and C81E in the BLI assay correlated with impaired ability to monomerise IRE1β LD in the SEC assay (Fig. EV3A-C). The C81W variant, that lacked any repressive activity in cells, had no measurable effect on IRE1β LD's monomer–dimer equilibrium (Fig. EV3D). Interestingly, the monomeric E60A mutant promoted the appearance of a pool of monomeric IRE1β LD when present at high concentration, suggesting that it is functionally a weak mutation (Fig. EV3E).

To obtain further insight into AGR2's mechanism of action, we sought to monitor IRE1β LD's monomer–dimer transition in real time. To this end, a Bioluminescence Resonance Energy Transfer (BRET)-based assay with recombinantly expressed and purified proteins was established. The assay exploits energy transfer from a nano luciferase (nluc) donor to a monomeric Green Lantern (mGL)

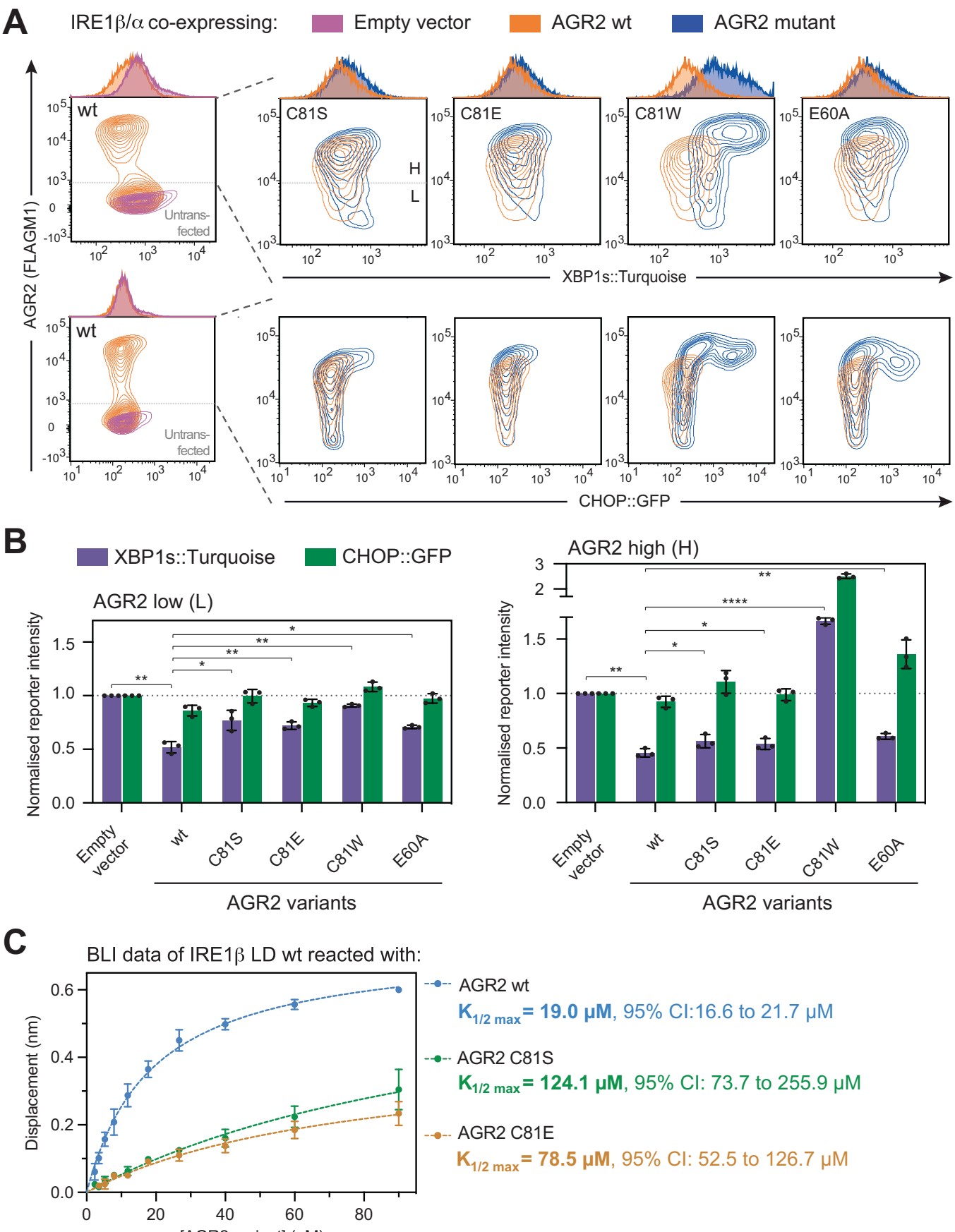

**Figure 5.  Active-site mutations impair AGR2 regulation of IRE1β's activity in cells and IRE1β LD binding in vitro.**

(A) Two-dimensional contour plot of AGR2 level (intracellularly FLAG-M1-stained) versus XBP1s::Turquoise or CHOP::GFP signals from dual UPR reporter IRE1β/α cells transiently transfected with the indicated AGR2 variants. Left panels show both, untransfected and transfected (FLAG-M1 positive) populations, whereas panels on the right were gated on the latter. (B) Quantification of reporter signals from (A) in AGR low (L) and AGR high (H) populations. Shown are the mean ± SD of three independent repetitions. Statistical analysis was performed by two-sided unpaired Welch's t test and significance is indicated by asterisks (*$P < 0.05$, **$P < 0.01$, ***$P < 0.001$, ****$P < 0.0001$). (C) Bio-Layer Interferometry (BLI)-derived signal from an IRE1β LD probe as a function of the concentration of the indicated interacting AGR2 proteins fitted to a one-site specific binding function with the indicated confidence intervals (CI). Shown are the mean ± SD of three independent repetitions. Source data are available online for this figure.

acceptor fluorophore, both fused to the flexible C-terminus of different IRE1β LD molecules (Fig. 6A). Alone, the IRE1β-nluc donor emission spectrum showed a single peak. The addition of IRE1β LD-mGL acceptor quenched the donor peak and resulted in a second acceptor emission peak (Fig. 6B). No BRET signal was observed by combining IRE1β-nluc donor with the same concentration of a free mGL acceptor (Fig. EV4A). Thus, BRET intensity tracked IRE1β LD dimerisation.

The addition of AGR2 to the IRE1β LD BRET pair resulted in a time-dependent decline in BRET and the establishment of a new steady state with lower BRET (Fig. EV4B), consistent with the notion that AGR2 increases the proportion of monomeric IRE1β LD. The BRET ratio at the new AGR2-induced steady-state reflects the proportion between BRET-competent dimers and BRET incompetent IRE1β LD configurations (e.g., monomers bound by AGR2). When plotted against the AGR2 concentration, the trace provides a measure of AGR2 action at equilibrium (Fig. 6C). Whilst reaching a similarly low BRET plateau at high enough concentrations, the C81S and C81E mutants had a higher $K_{1/2\ max}$ (compared to wild-type AGR2) in this metric of monomerisation (Figs. 6C and EV4B). The AGR2 E60A monomeric mutant displayed similar features to wild-type AGR2 in this equilibrium analysis. These findings are consistent with a model whereby AGR2 binds and stabilises the IRE1β LD monomer, decreasing in effect the 'on rate' of dimerisation (Fig. EV4C).

The rate at which the BRET measurement (in the presence of different concentrations of AGR2) approached its new equilibrium hinted at an additional component to AGR2 action: Challenging the IRE1β LD BRET pair with increasing concentration of unlabelled IRE1β LD competitor, led to a concentration-dependent decrease in the equilibrium BRET signal (Fig. EV4D) but the rate of change was independent of unlabelled competitor concentration (Figs. 6D, left panel and EV4E). This is expected, as unlabelled IRE1β LD functions exclusively as a labelled-monomer stabilising agent. Therefore, the rate at which the BRET changed in this experiment is predicted to be dominated by the dissociation of the IRE1β LD BRET-competent dimer and to be unaffected by the competitor. By contrast, upon addition of wild-type AGR2 we observed a concentration-dependent acceleration of the rate at which BRET approached its new, lower equilibrium values (Fig. 6D, right panel, E). These observations hint at a component of dimer destabilisation driven by wild-type AGR2, effectively increasing the 'off-rate' of the IRE1β LD dimer (Fig. EV4C). Interestingly, this feature was attenuated by active-site AGR2 mutations.

## The IRE1β-AGR2 pair responds to AGR2's client, MUC2

AGR2 repressed basal IRE1β/α signalling to levels below those observed in IRE1α/α expressing cells. However, despite the headspace

thus created for activation, the AGR2-repressed IRE1β/α had an attenuated response to ER stress-inducing drugs (tunicamycin or thapsigargin) that strongly activated both IRE1α/α and the parallel CHOP::GFP reporter (Fig. 7A). This reduced responsiveness in IRE1β/α activation was dependent on AGR2 abundance. Compared to retroviral transduction, AGR2 complementation as a transgene with a high-level expression promoter further reduced IRE1β/α's sensitivity towards pharmacologically induced ER stress (Fig. 7B). Furthermore, whereas introduction of SubA, a protease that inactivates BiP by cleaving its interdomain linker (Paton et al, 2006), strongly activated both IRE1α/α signalling and the PERK-dependent CHOP::GFP reporter, IRE1β/α appeared less responsive to BiP depletion (Fig. EV5A). Together, these observations suggest that the IRE1β-AGR2 couple may have acquired sensitivity to different signals than the IRE1α-BiP couple.

To examine this possibility, we turned to a major client of AGR2, MUC2, whose AGR2-dependent maturation relies on the formation of intra- and interchain disulphide bonds. Introducing the cysteine-rich N-terminal part of MUC2 into the ER of IRE1β/α; AGR2+ cells resulted in de-repression of IRE1 activity correlating with N-MUC2 abundance (Fig. 7C). When expressed at very high levels, N-MUC2 perturbed ER proteostasis as indicated by high CHOP::GFP levels. Therefore, to gate on IRE1β/α-selective effects, only the MUC2 low population was considered for quantification (Fig. 7C, gate marked with L). De-repression of the IRE1β/α-AGR2 pair was MUC2 length dependent, as it was only observed upon expression of the long version of N-MUC2 (residues 21–1397); a shorter version (21–1259) remained without major effect (Fig. 7D), though both were expressed at similar levels (Fig. EV5B). Expression of the cysteine-rich C-terminal part of MUC2 (4198–6178) also de-repressed the IRE1β/α-AGR2 pair. Importantly, low-level expression of the various MUC2 truncations had a weaker effect on IRE1α/α activity. These findings suggest that the IRE1β-AGR2 pair is sensitised to MUC2 whilst the IRE1α-BiP is more responsive to the toxins tunicamycin or thapsigargin that generally perturb ER proteostasis.

To obtain a physiological perspective on the reversible repressive role of AGR2, played out in CHO reporter cells artificially expressing an IRE1β-AGR2 pair, we compared the concentration of AGR2 in samples derived from mouse colonic lysates and IRE1β/α expressing CHO cells by quantitative immunoblotting (using known quantities of purified bacterially expressed AGR2 to calibrate the assay, Fig. EV5C). The estimated concentration of AGR2 in the ER of goblet cells (460 μM) is in the range of the IRE1α repressor BiP [~300 μM measured in pancreatic cells (Wang et al, 2019)] and is in fact higher than the concentration attained even in the highest IRE1β/α AGR2 expressing reporter cells (140 μM). There fore, though the interplay of MUC2, AGR2 and IRE1β LD is enacted here in a

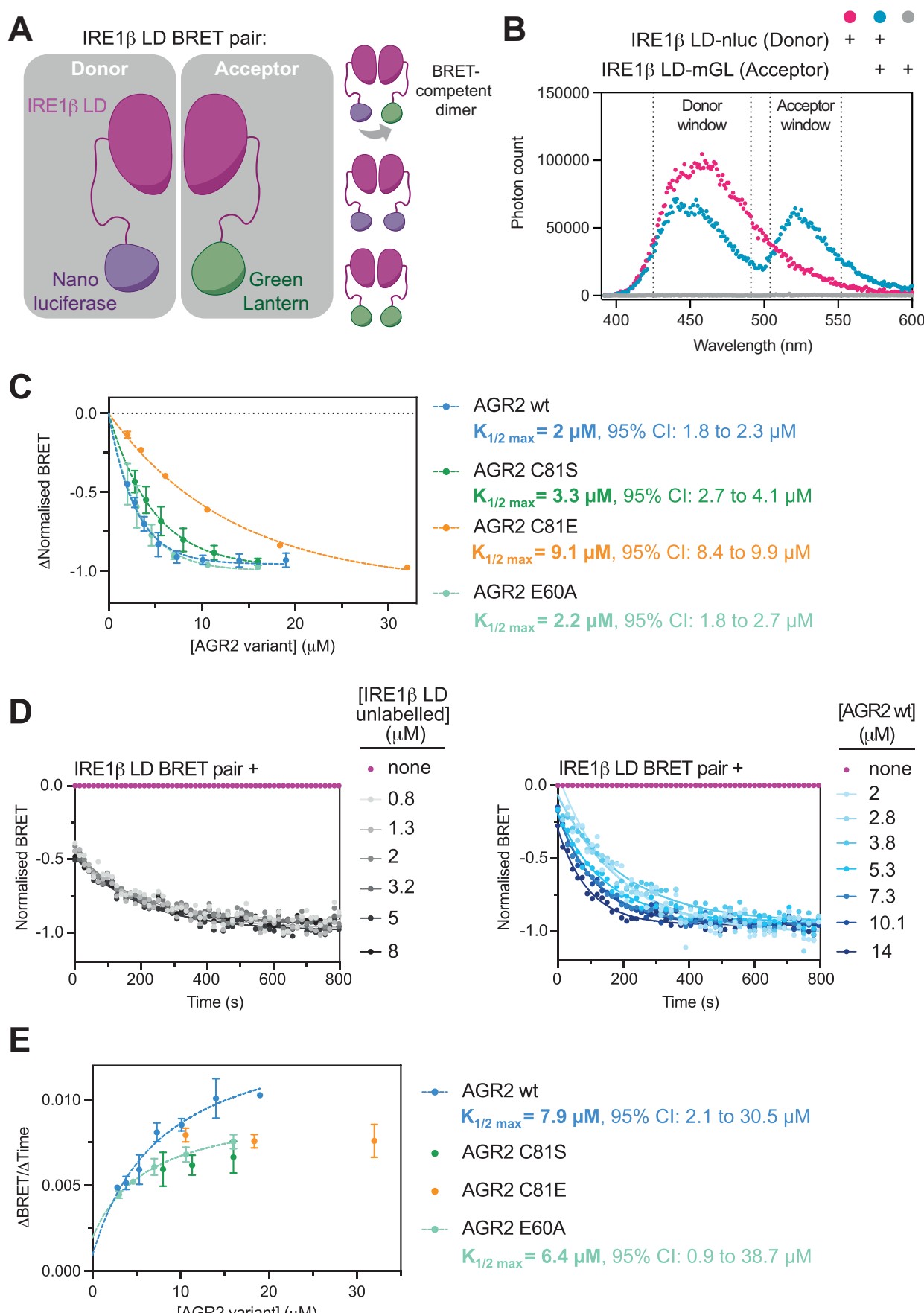

**Figure 6. Dimer destabilisation contributes to AGR2-driven monomerisation of IRE1β luminal domain (LD).**

(A) Schema of the IRE1β LD pair in the bioluminescence resonance energy transfer (BRET) assay. IRE1β LD was fused to a nano luciferase (nluc) donor or a monomeric Green Lantern (mGL) acceptor. (B) Plot of spectral scan showing the photon count per wavelength of the indicated samples. BRET intensity was computed based on measurements within the marked windows (425–491 nm and 480–528 nm, respectively). (C) Plot of the difference in BRET (baseline minus AGR-induced plateau, normalised ΔBRET) as a function of AGR2 concentration fitted to a one-site binding function with the indicated confidence intervals (CI). Shown are the mean ± SD of three independent repetitions. (D) Plot of time-dependent change in normalised IRE1β LD dimerisation-induced BRET following introduction of an unlabelled IRE1β LD competitor (left) or wild-type (wt) AGR2 (right). Traces were fitted to a one-phase exponential decay. (E) Plot of rate of BRET decrease as a function of AGR2 concentration fitted to a one-site binding function with a background value. The background is expected to be the 'off-rate' of the IRE1β LD dimer determined by addition of an unlabelled competitor (Fig. EV4E) as AGR2 concentrations approach zero. Shown are the mean ± SD of three independent repetitions. Representative primary kinetic data are found in Fig. EV4B,D. The $K_{1/2}$ max with 95% CI's is provided for wt and the E60A mutant, whereas no fit was obtained for C81S and C81E mutants. Source data are available online for this figure.

heterologous system, its outcome is unlikely to merely reflect trivial over-expression artefacts.

Overexpressed BiP repressed both, active IRE1β/α (in IRE1β/α expressing cells) and IRE1α/α (upon ER stress induction) as indicated by reduced XBP1s::Turquoise reporter levels (Fig. EV5D). Whilst AGR2 predominantly serves as a repressor for IRE1β, these findings suggest that BiP is likely to partially contribute to its regulation as well.

## IRE1β specialisation extends to its kinase–endonuclease extension domain

The IRE1β/α chimera, described above, enabled a reductionistic exploration of the stress-sensing properties of the IRE1β LD. To determine if IRE1β's functional specialisation extends to its cytosolically localised effector, we turned to a different domain-swap strategy that uncoupled stress-sensing from effector function: chimeric IRE1 proteins with variable α or β cytosolic kinase–endonuclease extension (KEN) effector portions, expressed from the endogenous *ERN1* locus.

A CRISPR-Cas9-mediated genomic deletion encompassing the endogenous IRE1α KEN domain [encoded by *ERN1* exons 12–22] abolished all XBP1 splicing and RIDD activity, as expected (Appendix Fig. S2A,B). Cell lines expressing IRE1α/α and IRE1α/β were obtained by CRISPR-Cas9-mediated homologous recombination of the deleted endogenous IRE1α ΔKEN locus with α or β repair templates (Fig. 8A). IRE1α/α and IRE1α/β were expressed at similar levels to the endogenous IRE1α (Appendix Fig. S2C and 'Methods').

Both chimeras restored basal and stress-induced XBP1 splicing. Splicing peaked at slightly lower levels in the IRE1α/β cells, whilst in wild-type (parental) cells and IRE1α/α cells, nearly all the unspliced XBP1 had been similarly depleted 6 h after ER stress induction by tunicamycin (Fig. 8B; Appendix Fig. S2D). Considering the similarity in protein expression levels (Appendix Fig. S2C), these observations suggest that IRE1α KEN has a higher XBP1 mRNA splicing activity than IRE1β KEN.

RIDD activity of the chimeric IRE1 proteins was estimated by measuring the level of five different known RIDD target mRNAs by qPCR (Hollien et al, 2009). To account for the impact of basal IRE1 activity on mRNA levels, we referenced RIDD to the mRNA levels of cells exposed to the IRE1 inhibitor 4μ8c (Cross et al, 2012). The presence of either chimera restored RIDD of the selected targets to similar levels (Fig. 8C,D). A two-dimensional plot of XBP1 splicing and RIDD suggests a slight bias in favour of RIDD over XBP1 splicing in the stressed IRE1α/β

cells (Fig. 8E; Appendix Fig. S2E). Together, these observations support a measure of specialisation of IRE1β's effector KEN domain, albeit less conspicuously than the specialisation of its stress-sensing LD.

## Discussion

The stress-sensing luminal domain (LD) couples IRE1 activity to changing conditions in the ER. Domain-swap experiments indicate that in a heterologous system lacking components present in IRE1β-expressing cells, the IRE1β LD is constitutively active. This finding points to the lack of an endogenous repressor of the IRE1β LD in the heterologous system and to the dominance of repression (by ER luminal agents) over activation (by ER ligands) in regulating IRE1β activity. This conclusion is bolstered by the identification of AGR2 as a repressor of IRE1β LD that is normally co-expressed in goblet cells and by the reconstruction of key facets of the repressive process in vitro. The findings presented here thus argue that direct repression by ER luminal chaperones is not merely a process that can regulate IRE1 activity in experimental setups but one with the potential to dominate the physiological regulation of an IRE1 isoform.

Caveats apply. Whilst it is formally possible that a conformational perturbation contributes to the constitutive activity of the IRE1β/α chimera, the accompanying manuscript (accompanying paper Cloots et al, 2023) shows that the intact IRE1β is also constitutively active when expressed in cells lacking AGR2. Could the constitutive activity of the IRE1β/α chimera in CHO cells reflect a corrupt concentration regime imposed by the genetic swap? In goblet cells, IRE1β mRNAs are ~50-fold more abundant than IRE1α mRNAs, whereas IRE1α expression, measured against housekeeping genes, is similar to that of a 'typical' mammalian cell (Haber et al, 2017). Thus, unphysiological high concentrations are unlikely to drive IRE1β LD activity in domain-swapped CHO cells in which the IRE1β/α chimera is expressed from the endogenous *ERN1* locus. It is formally possible that CHO cells (that do not express IRE1β endogenously) happen to be especially enriched in IRE1β LD activating ligand(s), and that AGR2, which is also normally missing from these cells, happens to be a powerful repressor of such ligands. Though not impossible, such a scenario seems implausible and is countered by other observations: (1) The physiological pairing of IRE1β with AGR2 is reflected in the phenotypic overlap in the consequences of their deficiency in mice (Bertolotti et al, 2001; Park et al, 2009; Zhao et al, 2010), (2) Endogenous IRE1β and AGR2 form a complex in mouse tissues

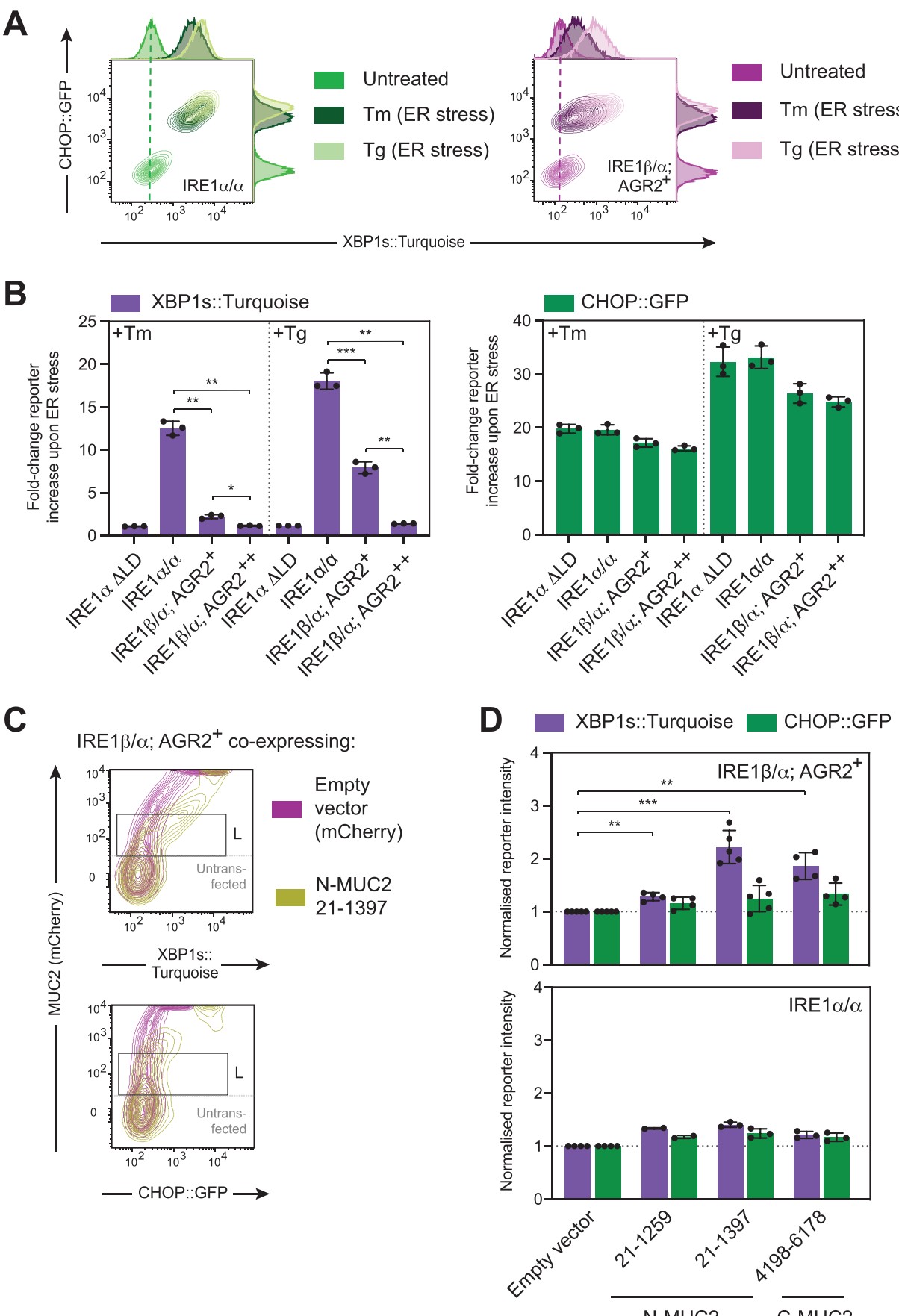

◄ **Figure 7. The IRE1β-AGR2 pair is selectively sensitive to titration by MUC2.**

(A) Two-dimensional contour plots of CHOP::GFP and XBP1s::Turquoise signals from dual UPR reporter IRE1α/α or IRE1β/α CHO cells, the latter stably expressing the AGR2 repressor. Cells were treated with the ER stressors tunicamycin (Tm) or thapsigargin (Tg). A representative data set out of three independent experiments is shown. (B) Quantification of fold-change XBP1s::Turquoise and CHOP::GFP signals upon ER stress induction of cells with the indicated genotype. IRE1β/α; AGR2++ cells express higher AGR2 levels than IRE1β/α; AGR2+ cells (see Fig. EV5B). Shown are the mean ± SD of three independent repetitions. Statistical analysis was performed by two-sided unpaired Welch's *t* test and significance is indicated by asterisks (*$P < 0.05$, **$P < 0.01$, ***$P < 0.001$). (C) Contour plots of CHOP::GFP and XBP1s::Turquoise signals of IRE1β/α; AGR2+ cells from (A) transiently transfected with cysteine-rich N-terminal MUC2 fragment marked with mCherry as a fiduciary of expression level. (D) As in (B), upper panel: Signals of IRE1β/α; AGR2+ cells transiently transfected with the indicated MUC2 variants gating on the low expressing population [gate marked with L in (C)]. The lower panel reports on the same measurements in IRE1α/α cells. Shown are the mean ± SD of three independent repetitions. Statistical analysis was performed by two-sided unpaired Welch's *t* test and significance is indicated by asterisks (**$P < 0.01$, ***$P < 0.001$). Source data are available online for this figure.

and endogenous AGR2 represses endogenous IRE1β in mucin-producing LS174T cells (accompanying paper Cloots et al, 2023), (3) AGR2 represses the activity of IRE1β/α in a reconstituted heterologous system, reported here 4) AGR2 antagonises IRE1β LD dimerisation in vitro.

Given the above, we favour a scenario whereby physiological regulation of IRE1β is subordinate to the occupancy of AGR2 by clients, such as mucins (Fig. 9). The limited scope of IRE1β/α activation by toxins that generally perturb ER function together with the limited scope for mucin expression to activate IRE1α and PERK, observed here, suggests that the specialised IRE1β-AGR2-mucin triple arose to solve the problem of matching activity of the IRE1 branch of the UPR to the load of a goblet cell ER client that is poorly detected by the non-specialised transducers. This speculation fits with the greater divergence in sequence of the LDs of the IRE1 isoforms over the divergence of their KEN domains and with our observation that the intrinsic effector activities of the two isoforms are rather similar.

The IRE1β-AGR2 and IRE1α-BiP couples share certain biochemical features: In both, the presence of chaperone disfavours the oligomeric, active state of IRE1, and in both the action of the chaperone is sensitive to the functional state of its active site. This is reflected in the dependence of BiP's ability to repress IRE1 signalling on its co-chaperone-stimulated ATPase activity (Amin-Wetzel et al, 2019, 2017) and in the sensitivity of AGR2 repression of IRE1β to mutations in its PDI active site. In both couples, the biophysical data suggests two potentially related processes: Active destabilisation of the IRE1 LD dimer-gleaned from the enhanced rate of dimer dissociation in the presence of the active chaperone. A role for chaperone binding in stabilising the monomeric LD-gleaned from the effect of chaperone on the LD monomer–dimer equilibrium and from features of the complex formed between the LD and the chaperone.

It is easy to imagine how an ATP-consuming machine could work to destabilise an IRE1α dimer/oligomer. However, AGR2 lacks ATPase activity. It has the potential to form mixed disulphides and thus exploit that as a source of chemical energy to destabilise the IRE1β LD dimer. Our findings cannot exclude a contribution of this mechanism to IRE1β repression observed in vivo, however, in vitro destabilisation of the dimer occurs under reducing conditions (that disfavour electron exchange with AGR2) and is also observed in an IRE1β LD preparation lacking the two LD cysteines. Moreover, the accompanying paper by Cloots et al, 2023 identified AGR2 as an interactor of IRE1β in an immuno-affinity screen performed under reducing conditions. Therefore, the sensitivity of AGR2 to mutations in its active-site

cysteine are better explained by the compromise of an allosteric signal arising from its PDI active site and affecting its chaperone activity, as has been demonstrated in other PDI family members (Serve et al, 2010; Wang et al, 2012).

Mucin load-dependent activation of IRE1β might proceed through titration of AGR2 as it forms mixed disulphides with its client. However, the difficulty in detecting mixed AGR2-mucin disulphides argues that these are transient species with limited scope for sequestering AGR2 (Park et al, 2009; Bergström et al, 2014). Alternatively, clients such as mucin may compete for the disulphide-independent client-binding activity of AGR2 (its conventional chaperone activity). Whilst mass action by a pool of AGR2 mixed disulphides seems unlikely, allosteric regulation of AGR2 by thiol-mediated exchange with protein substrates (or other redox regulators) is plausible. Furthermore, regulators of AGR2 dimerisation have been reported to influence its role in inflammation (Maurel et al, 2019). This suggests an additional mode for regulating IRE1β activity by AGR2 and hints that such regulation may go beyond simple titration of an abundant chaperone by abundant clients. Our analysis of the monomeric AGR2 E60A mutant remains incomplete: whilst it retains some ability to repress IRE1β/α signalling in cells and can promote a pool of monomeric IRE1β LD in vitro, irreversible binding to the BLI probe preclude correlating these activities with binding to IRE1β LD.

Here we emphasise the evidence for IRE1 specialisation at the level of its LD and the prospects of a simple mechanism of regulation, based on titration of a repressing chaperone. However, none of this argues against additional functional specialisation in IRE1β, for example, at the level of its effector outputs. Whilst the IRE1α/β chimera used here to isolate the effector functions of IRE1β from its upstream regulation point to only modest differences in the intrinsic RIDD and XBP1 splicing activity of the two IRE1 isoforms, it is impossible to rule out a distortion imposed on the measurements by the chimera. Furthermore, a preference for RIDD, which may regulate the abundance of mRNAs encoding secreted proteins such as mucin (Tsuru et al, 2013; Nakamura et al, 2011) or link IRE1β activity to cell death [(Iwawaki et al, 2001) and accompanying manuscript by Cloots et al, 2023], could be driven by high concentrations of IRE1β in goblet cells or by specialisation in target selection (both would be missed in the CHO cells studied here).

A recent examination of the oligomeric state of endogenous IRE1α indicates that activation in vivo hinges on the transition from dimers to tetramers (Belyy et al, 2022), a finding that agrees with the existence of two homotypic interaction surfaces on IRE1α's LD

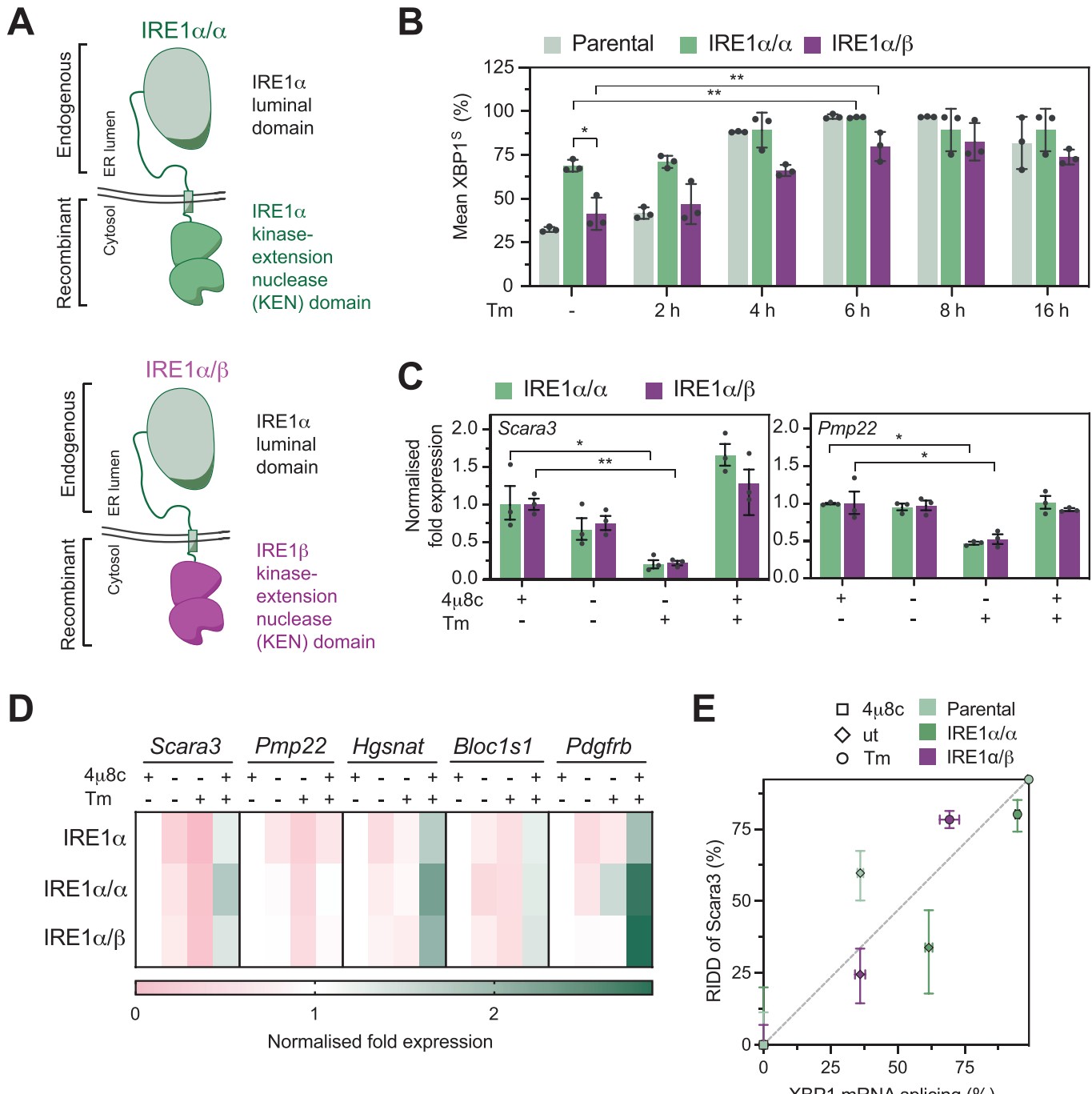

(Karagöz et al, 2017). By contrast, we find that purified IRE1β LD in solution is distributed in particles whose mass corresponds to monomers and dimers. These observations are consistent with the SEC elution profile of detergent-solubilised full-length IRE1β isolated from mammalian cells (Grey et al, 2020) and suggest that the coupling between oligomerisation and activation may be different for the two isoforms of IRE1.

AGR2's repressive effect is selective to the beta isoform. Whilst we detected only a minimal effect on IRE1α signalling in cells, it failed to bind and monomerise IRE1α LD. BiP binding, however, is potentially more promiscuous, leaving room for a residual role for

BiP in repressing IRE1β. This is hinted at by the observation that depletion of BiP by the SubA protease induced signalling of IRE1β/α in the presence of AGR2. Moreover, overexpressed BiP repressed the constitutively active IRE1β/α. This should come as no surprise, given that the gene duplication that gave rise to the two isoforms occurred in an organism with an existing IRE1-BiP couple. Residual BiP repression of IRE1β may arise directly, as suggested by findings that the two proteins can interact (Bertolotti et al, 2000), or indirectly by competing with putative activating ligands (Oikawa et al, 2012). Details of the crosstalk between BiP and IRE1β remain to be addressed.

**Figure 8.   A mild bias towards RIDD over XBP1 mRNA splicing by IRE1β's kinase–endonuclease extension (KEN) domain in cells.**

(A) Schematic representation of the chimeric IRE1 variants that were used in this study. IRE1α/α and IRE1α/β, comprising the endogenous IRE1α luminal domain and variable mouse KEN domain, were created by CRISPR/Cas9-mediated knock-in into IRE1α ΔKEN cells. (B) Plot displaying the level of spliced XBP1 (XBP1$^S$) in cells with indicated genotype. XBP1 mRNA splicing was assessed by RT-PCR and gel electrophoresis. The percentage of XBP1$^S$ is given relative to total XBP1 mRNA. Cells were treated with Tunicamycin (Tm) as indicated. The bars and error bars represent the mean ± SEM of data obtained from three independent experiments. Statistical analysis was performed by two-sided unpaired Welch's $t$ test and significance is indicated by asterisks (*$P < 0.05$, **$P < 0.01$). (C) Plot of the normalised fold expression of RIDD targets Scara3 and Pmp22 in IRE1α/α and IRE1α/β expressing cells as determined by qPCR. The cells were treated with Tm or the IRE1 inhibitor 4μ8c (Cross et al, 2012) as indicated. mRNA levels referenced to cells in which (basal) IRE1 activity had been blocked by 4μ8c. The bars and error bars represent the geometric mean ± SEM of data obtained from three independent experiments. Statistical analysis was performed by two-sided unpaired Welch's $t$ test and significance is indicated by asterisks (*$P < 0.05$, **$P < 0.01$). (D) Colour-coded heat map showing relative expression of RIDD targets in cells of the indicated genotypes. Experimental set up as described in (C). (E) Two-dimensional plot of Scara3-directed RIDD and XBP1 mRNA splicing activity of inhibited (4μ8c), untreated (ut) and stressed (8 h Tm) parental, IRE1α/α and IRE1α/β CHO cells. Data points plotted are taken from qPCR data in (C) and RT-PCR data in Appendix Fig. 2D. Source data are available online for this figure.

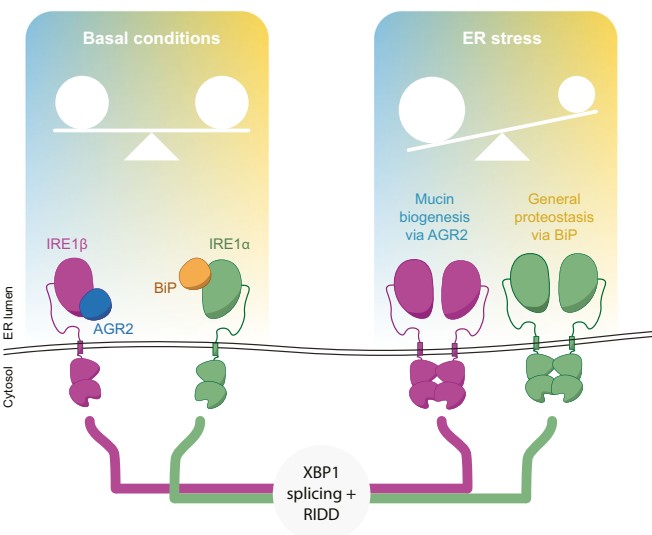

**Figure 9.   Schema contrasting IRE1α and IRE1β regulation.**

Both isoforms are controlled by titratable repressors, imposing an inactive monomeric state on the UPR transducer. BiP, a chaperone with a broad clientele, couples the widely expressed IRE1α to general protein-folding homoeostasis in the ER. The gene duplication that gave rise to IRE1β established an isoform that is coupled narrowly to the specialised clientele of the ER of goblet cells by AGR2, a mucin-selective chaperone. This arrangement solved the problem arising from the relative insensitivity of the IRE1α-BiP couple to mucin and enables goblet cells to match XBP1 signalling (and its downstream effectors) to the burden imposed by its specialised clientele.

The mechanistic issues noted above remain to be resolved. Nonetheless, the intriguing discovery of a cell type-specific specialisation of the UPR reported here (and in the accompanying paper by Cloots et al, 2023) and insights it provides to general principles regulating UPR transducers will hopefully fuel further research into this interesting problem of cell physiology.

## Methods

The plasmids used in this study are listed in Table 1. Primer pairs used in this study and their primer efficiency (E) are listed in Table 2.

## Mammalian cell culture

CHO-K1 cells (ATCC CCL-61) were phenotypically validated as proline auxotrophs, and their *Cricetulus griseus* origin was confirmed by genomic sequencing. CHOP::GFP and XBP1s::Turquoise reporters were introduced sequentially under G418 and puromycin selection to generate the previously described derivative CHO-K1 S21 clone (Sekine et al, 2015). Cell lines were subjected to random testing for mycoplasma contamination using the MycoAlert Mycoplasma Detection Kit (Lonza).

For IRE1 domain swaps, see 'Gene manipulation and allele analysis'. Full-length IRE1β and AGR2 over-expression was performed with CHO-K1 cells (ATCC CCL-61) (Fig. 3C). Cells were cultured in Ham's nutrient mixture F12 (Sigma). All cell media was supplemented with 10% (v/v) serum (FetalClone-2, Hyclone), 2 mM L-glutamine (Sigma), 100 U/ml penicillin and 100 μg/ml streptomycin (Sigma). HEK293T cells (ATCC CRL-3216) were cultured in Dulbecco's modified Eagle's medium (Sigma) supplemented as described above. Cells were grown in tissue culture dishes or multi-well plates (Corning) at 37 °C and 5% $CO_2$. Where indicated, cells were treated with Tunicamycin (Melford) at 2.5 μg/ml for the indicated time or 16 h, 2-deoxyglucose (2DG) (Sigma) 4 mM for 16 h and 4μ8c (Cross et al, 2012) at 16 μM for 3 days. The drugs were mixed with pre-warmed culture medium and immediately added to the cells by medium exchange.

## Transfection

Cells were transfected using Lipofectamine LTX (Life Technologies) transfection reagent with reduced serum medium Opti-MEM (Life Technologies) following the manufacturer's instructions. In Figs. 2B–D, EV1B and EV5A,D, cells were analysed 72 h after transfection. In Figs. 3C, 5A and 7C, cells were harvested 48 h after transfection. To genetically complement IRE1β/α expressing dual UPR reporter cells with FLAG-M1-tagged AGR2, retroviral transduction was performed to create an IRE1β/α; AGR2$^+$ cell line. For this, HEK cells were transfected with TransIT-293 Transfection Reagent (Mirus) according to the manufacturer's instructions (see 'Genetic complementation of AGR2' for details).

## Gene manipulation and allele analysis

For IRE1 LD swaps, we used the previously described ΔLD15 dual CHOP::GFP and XBP1s::Turquoise UPR reporter CHO-K1 cell

**Table 1.** Plasmids used in this study.

| Unique | Plasmid name | Figure | Reference | Description |
|---|---|---|---|---|
| UK1903 | CHO_IRE1_guideC15.1_pSpCas9(BB)-2A-mCherry | 1B | Kono et al, 2017 | Cas9 and guide targeting IRE1 in CHO-K1 ΔLD clone 15 (mCherry-tagged) |
| UK1968 | CHO_IRE1_hIRE1-LD_reptemp4_pCR-Blunt2-TOPO | 1B | Kono et al, 2017 | Repair template for wild-type hIRE1a LD reconstitution in CHO-K1 cells |
| UK2757 | CHO_ mIRE1b_LD_39-426_IRE1a_3xFLAG_reptemp4 | 1B, EV1A | This study | Repair template for wild-type mIRE1b LD reconstitution in CHO-K1 cells |
| UK2881 | pBABEpu_FLAGM1_mAGR2 | 2A | This study | Plasmid used for retroviral transduction of FLAG-M1 AGR2 |
| UK40 | pL_VSVG | 2A | Bartz and Vodicka, 1997 | CMV_VSVG (RV packaging), for retroviral transduction of FLAG-M1 AGR2 |
| UK41 | pJK3 | 2A | Bartz and Vodicka, 1997 | GAG-POL, for retroviral transduction of FLAG-M1 AGR2 |
| UK42 | pCMV_TAT_HIV | 2A | Bartz and Vodicka, 1997 | Transactivator, for retroviral transduction of FLAG-M1 AGR2 |
| UK1314 | pCEFL_mCherry_3XFLAG_C | 2B,D, 3C, EV1B, 7C,D, EV5A,B | Sekine et al, 2015 | pCEFL with 3XFLAG_C tagged from mCherry-tagged plasmid, empty vector of UK2708 |
| UK2708 | muAGR2_pCEFL_mCherry_FLAG_M1 | 2B,D, 3C, EV1B | This study | Mammalian expression of FLAG-M1-tagged muAGR2 from mCherry-tagged plasmid |
| UK2709 | muAGR2_pCEFL_mCherry | 2B | This study | Mammalian expression of muAGR2 from mCherry-tagged plasmid |
| UK1610 | pSpCas9(BB)-2A-mCherry_V2 | 2E | Amin-Wetzel et al, 2017 | Modified pSpCas9(BB)-2A vector to express mCherry together with guide RNA & Cas9 |
| UK2771 | FLAGM1_muAGR2_guide1_pSpCas9(BB)-2A-mCherry_V2_MP2 | 2E | This study | pSpCas9(BB)-2A vector to express mCherry together with guide RNA & Cas9 to knock out FLAG-M1-tagged muAGR2 |
| UK2772 | FLAGM1_muAGR2_guide2_pSpCas9(BB)-2A-mCherry_V2_MP5 | 2E | This study | pSpCas9(BB)-2A vector to express mCherry together with guide RNA & Cas9 to knock out FLAG-M1-tagged muAGR2 |
| UK2246 | IRE1a_LD_ΔC_24-443_AviTag_H6_pET30a(+) | 3A | Amin-Wetzel et al, 2019 | C-terminally-tagged AviTag-H6 human IRE1a LD ΔC |
| UK2796 | pMAL-hIRE1bNLD_35-428_AT_His7 | 3A, 5C, 6D, EV4D | This study (hIRE1b LD originating from Oikawa et al, 2012 | Bacterial expression MBP-hIRE1b_LD_(35-428)-AviTag-His7 |
| UK1107 | mIRE1b_pCDNA5_FRT_TO | 3C | This study | Full-length mIRE1b |
| UK916 | pCDNA5_FRT_TO | 3C | This study | Empty vector for UK1107 |
| UK2758 | CHO_ mIRE1b_LD_39-426_dC_IRE1a_3xFLAG_reptemp4 | EV1A | This study | Repair template for mIRE1b LD dC reconstitution in CHO-K1 cells |
| UK3028 | pMAL-hIRE1bNLD_ΔC-mGL-AT-His | EV1C | This study | Bacterial expression hIRE1b-ΔC version of UK2986 |
| UK2986 | pMAL-hIRE1bNLD-mGL-AT-His | 4A-D, EV2A-C,E, EV3A-E, 6B-E, EV4A,B,D,E | This study | C-term mGL inserted in MBP-hIRE1bLD in UK2796 |
| UK2794 | H6pSUMO3_mAGR2_21_175 | 4A, B, D, EV2C-E, 5C, EV3A, 6C-E, EV4B, EV5C | This study | Bacterial expression of H6pSUMO3_mAGR2 WT |
| UK2048 | pET22b_H7_Smt3_Ire1a_LDΔC_24_444 R234C | 4B | Amin-Wetzel et al, 2017 | Bacterial expression of Smt3-tagged cysteine-free IRE1a LDΔC R234C, for labelling with Oregon Green |
| UK3053 | pMAL-hIRE1bNLD_35-428_ΔC_AT_His7 | EV2C-E | This study | Bacterial expression hIRE1b-ΔC version of 2796 |
| UK13 | pCEFL.puro | 5A | Sekine et al, 2015 | Empty vector of UK2718 |

**Table 1.** (continued)

| Unique | Plasmid name | Figure | Reference | Description |
|---|---|---|---|---|
| UK2718 | SP_FLAGM1_muAGR2_pCEFL-puro_constr_MP1 | 5A, EV5C | This study | Mammalian expression of FLAG-M1-tagged muAGR2 with puro resistance for intracellular staining and insertion of AGR2 as a transgene |
| UK2865 | SP_FLAGM1_muAGR2_C81S_pCEFL-puro_MP2 | 5A | This study | Mammalian expression of muAGR2 C81S with puro resistance for intracellular staining |
| UK2866 | SP_FLAGM1_muAGR2_E60A_pCEFL-puro_MP5 | 5A | This study | Mammalian expression of muAGR2 E60A with puro resistance for intracellular staining |
| UK2871 | SP_FLAGM1_muAGR2_C81W_pCEFL-puro_MP6 | 5A | This study | Mammalian expression of muAGR2 C81W with puro resistance for intracellular staining |
| UK2873 | SP_FLAGM1_muAGR2_C81E_pCEFL-puro_MP10 | 5A | This study | Mammalian expression of muAGR2 C81E with puro resistance for intracellular staining |
| UK2795 | H6pSUMO3_mAGR2_21_175_C81S | 5C, EV3B, 6C,E | This study | Bacterial expression of H6pSUMO3_mAGR2 C81S |
| UK2835 | H6pSUMO3_mAGR2_C81E_21_175 | 5C, EV3C, 6C,E, EV4B | This study | Bacterial expression of H6pSUMO3_mAGR2 C81E |
| UK2833 | H6pSUMO3_mAGR2_C81W_21_175 | EV3D | This study | Bacterial expression of H6pSUMO3_mAGR2 C81W |
| UK2816 | H6pSUMO3_mAGR2_21_175_E60A | EV3E, 6C,E | This study | Bacterial expression of H6pSUMO3_mAGR2 E60A |
| UK3155 | pMAL-hIRE1bNLD_35-428_Nanoluc_H6 | 6B-E, EV4B,D,E | This study | NanoLuc (complete, WT)-tagged MBP-hIRE1bLD (35-428) bacterial expression vector, a BRET donor |
| UK3060 | H6_pSUMO3_MGL-MP4 | EV4A | This study | Bacterial expression of mGreen lantern |
| UK2805 | huMUC2_21-1259_pCEFL_mCherry (MP23) | 7D, EV5B | This study, MUC2 originating from Javitt et al, 2019 | Mammalian expression N-term of human MUC2 fragment (from Debbie Fass) |
| UK2804 | huMUC2_21-1397_pCEFL_mCherry (MP21) | 7C,D, EV5B | This study, MUC2 originating from Javitt et al, 2019 | Mammalian expression N-term of human MUC2 fragment (from Debbie Fass) |
| UK2947 | FLAGM1_huMUC2_4198-5179_pCEFL_mCherry_MP1 | 7D | This study, MUC2 originating from Lidell et al, 2003 | Mammalian expression FLAG-M1-tagged human MUC2 C-term. in pCEFL-mCherry, from Gunnar Hansson |
| UK2952 | cglRE1a_Int12_g2_pSpCas9(BB)-2A-mCherry | Appendix Fig. S2A-C | This study | CRISPR guide vector targeting intron 12 to create IRE1a ΔKEN deletion |
| UK2876 | cglRE1a_Ex22_g1_pSpCas9(BB)-2A-mCherry | Appendix Fig. S2A-C | This study | CRISPR guide vector targeting exon 22 to create IRE1a ΔKEN deletion |
| UK3022 | endogenous_5'HA_mIRE1b_Ken-bGHT_repair_V1 | 8B-E, Appendix Fig. S2D,E | This study | Repair template for IRE1b KEN reconstitution in CHO-K1 cells |
| UK3093 | endogenous_5'HA_mIRE1a_Ken_repair_bGHT_pBSKS | 8B-E, Appendix Fig. S2D,E | This study | Repair template for IRE1a KEN reconstitution in CHO-K1 cells |
| UK2668 | SENP2_364-589_pET-28a_MP9 | | This study | Bacterial expression of human SUMO3 protease SENP2 active fragment to cleave SUMO-tag |
| UK1452 | SubA_22-347_3xFLAG_KDEL_pCEFL_mCherry | EV5A | Amin-Wetzel et al, 2019 | Mammalian expression of SubA protease |
| UK1459 | SubA_22-347_S272A_3xFLAG_KDEL_pCEFL_mCherry | EV5A | Amin-Wetzel et al, 2019 | Mammalian expression of SubA protease mutant variant |
| UK 2557 | haBiP_pCEFL_mCherry | EV5D | This study | Mammalian expression of BiP |
| UK887 | hPDI1_WT_mCherry_KDEL | 2D | Avezov et al, 2015 | Mammalian expression of PDI1 |

**Table 2.  Primer pairs used in this study and their primer efficiency (E).**

| Gene | Primer | Sequence (5′ → 3′) | E (%) | Use |
|------|--------|--------------------|-------|-----|
| XBP1 | 1470 hamXBP1.19 S<br>5 mXBP1.14AS | GGCCTTGTAATTGAGAACCAGGAG<br>GAATGCCCAAAAGGATATCAGACTC | — | RT-PCR |
| Scara3 | 3237 cgScara3.1 S<br>3238 cgScara3.1AS | GGCTCTGCTCCTTGTGGCCG<br>CAGGGCTTTTGGGTCCAGTCCT | 95.2 | qPCR |
| Pmp22 | 3243 cgPmp22.1 S<br>3244 cgPmp22.1AS | TCGTCAGCGAGTGAATGGCTACA<br>GCTGCTGCACTCATCACGCA | 102.5 | qPCR |
| Hgsnat | 3235 cgHgsnat.1 S<br>3236 cgHgsnat.1AS | CTCCACCGTCCTTTATCACACCCAG<br>CACCAGGCAGTGAATCTCATCAGG | 107.0 | qPCR |
| Bloc1s1 | 52 MmBlos1.1 S<br>53 MmBlos1.2AS | CAAGGAGCTGCAGGAGAAGA<br>GCCTGGTTGAAGTTCTCCAC | 109.9 | qPCR |
| Pdgfrb | 3241 cgPdgfrb.2 S<br>3242 cgPdgfrb.2AS | AAACCCCCTACAGCTGTCTT<br>CAATCCCCATGGCGTCTGCG | 108.1 | qPCR |
| Rpl27 | 3249 cgRpl27.2 S<br>3250 cgRpl27.2AS | ACAATCACCTCATGCCCACAAG<br>GCGTTTCAGGGCTGGGTCTC | 105.3 | qPCR |
| GAPDH | 3255 cg.GAPDH.1 S<br>3256 cg.GAPDH.1AS | TTTCCGTGCAGTGCCAGCCT<br>CCAGGCGTCCAATACGGCCA | 106.8 | qPCR |

line (Kono et al, 2017) as the parental strain for the CRISPR-Cas9-mediated homologous recombination. The deleted LD domain was reconstituted by co-transfection of ΔLD15 cells with a CRISPR guide plasmid targeting the endogenous *ERN1* locus (UK1903) together with the respective repair templates, encoding either human (h) IRE1α LD (UK1968) or murine (m) IRE1β LD (UK2757 for wild-type, UK2758 for ΔC LD), in a 1:9 ratio. Integration of the sequences into the endogenous *ERN1* locus was confirmed by sequencing of the 5'-integration sites for both genotypes and by mass spectrometry for peptides mapping to either the IRE1α or IRE1β LD (Fig. 1E). For MS analysis, tryptic digests of material after immunoprecipitation was prepared using an iST 96x sample preparation kit (Preomics). In IRE1β ΔC/α cells, integration was confirmed by sequencing of the 5'-integration sites only.

For the IRE1 KEN domain swaps, CHO cell lines used in this study were created by CRISPR/Cas9-mediated knockout of the endogenous *Cricetulus griseus* IRE1α KEN domain of dual UPR reporter CHO-K1 cells (Sekine et al, 2015) followed by the CRISPR/Cas9-mediated knock-in of mIRE1α KEN or mIRE1β KEN encoding cDNA-derived minigene (lacking exons) into the endogenous *ERN1* locus. For the deletion of endogenous IRE1α KEN, CHO cells were co-transfected with a Cas9 encoding vector (UK1610) and CRISPR guide vectors targeting intron 12 and exon 22 of *ERN1* (UK2952, 2953, 2876) in a 1:1 ratio. The deletion was confirmed by sequencing. The resultant IRE1α ΔKEN cells retained the portion of *ERN1* encoding the luminal (1–450 aa) and transmembrane domain (451–473 aa). The deleted KEN domain was rescued by co-transfection of IRE1α ΔKEN cells with a CRISPR guide plasmid (UK2952) and a rescue allele plasmid, encoding either mIRE1α KEN (UK3093) or mIRE1β KEN (UK3022), in a 1:9 ratio. Knock-in resulted in the replacement of the endogenous 3' poly(A) and transcription termination signals with those of the repair plasmid-derived bovine growth hormone. Integration of the sequences into the endogenous *ERN1* locus was confirmed by sequencing of the 5'-integration sites for both genotypes and by mass spectrometry of the IRE1α antibody-reactive proteins in the IRE1α/β and IRE1α/α cells.

Cas9 guides were either manually designed following standard guidelines (Ran et al, 2013) or taken from the CRISPy database (URL: http://staff.biosustain.dtu.dk/laeb/crispy/; Ronda et al, 2014). Cells were transfected with the Cas9 and guide constructs and grown for seven days before they were analysed by flow cytometry or fluorescence-activated cell sorting (FACS). For CRISPR/Cas9-mediated knockout of the AGR2 transgene (Fig. 2D), cells were co-transfected with a Cas9 encoding vector (UK1610) and CRISPR guide vector (UK2771 or UK2772) in a 1:1 ratio and analysed six days after transfection.

Genomic DNA was extracted from final clones, PCR used to amplify the loci of interest and the resultant products were sequenced. The genomic DNA was extracted from cells grown to 90% confluency in a 12-well dish by incubation in 500 µl Proteinase K solution (100 mM Tris-HCl pH 8.5, 5 mM EDTA, 200 mM NaCl, 0.25% SDS, 0.2 mg/ml Proteinase K) overnight at 50 °C. Next, DNA was precipitated by adding 600 µl isopropanol and reactions rotated for 1 h at room temperature. Samples were spun 10,000×*g* for 15 min and the supernatant was carefully removed. To wash the pellet, 500 µl 70% Ethanol was added and samples were spun 10,000×*g* for 15 min. This washing step was performed twice. Pellets were dried for 30 min in a hood, resuspended in 500 µl sterile water and used as a template in PCR reactions before sequencing.

## Flow cytometry and fluorescence-activated cell sorting (FACS)

To analyse the effect of IRE1 variants expressed from the endogenous *ERN1* locus on the UPR (Figs. 1B,C, 2A–E, EV1A,B, 7A,C and EV5A,D), flow cytometry was performed. Cells were washed once in PBS and collected in PBS containing 4 mM EDTA. Single-cell fluorescent signals (20,000/sample) were analysed by multi-channel flow cytometry with an LSRFortessa cell analyser (BD Biosciences). CHOP::GFP fluorescence was detected with excitation laser at 488 nm, filter 530/30 nm; XBP1s::Turquoise fluorescence with excitation laser 405 nm, filter 450/50 nm, mCherry fluorescence with excitation laser 561, filter 610/20 and

FLAG-M1 intracellular staining at 640 nm, filter 730/45 nm. Data were processed using FlowJo, and median reporter analysis was performed using Prism 9 (GraphPad).

FACS was performed on either a Beckman Coulter MoFlo or a BD FACSMelody cell sorter. Cells were washed once in PBS and then incubated 5 min in PBS supplemented with 0.5% BSA and 4 mM EDTA before sorting into fresh media. To generate clonal cell lines stably expressing a version of IRE1 from the endogenous *ERN1* locus, the transfected cells were treated with 2DG (Sigma) to gate for cells showing high CHOP::GFP and XBP1s::Turquoise fluorescence.

## Intracellular FLAG-M1 staining of AGR2 variants for flow cytometry analysis

For intracellular staining of AGR2 variants (Fig. 5A), the protocol from (Preissler et al, 2020) was adapted. In brief, cells were grown in a six-well plate and analysed 48 h after transfection. For analysis, cells were washed once in PBS and collected in PBS containing 2 mM EDTA. After spinning for 6 min at $100 \times g$ at 4 °C, cells were fixed by adding 4% formaldehyde for 10 min whilst rotating. Next, cells were centrifuged at $2300 \times g$ for 5 min at room temperature, and the supernatant was discarded. Cells were resuspended in 500 μl of blocking/permeabilisation/wash solution (PBS, 0.1% Triton X-100, 10% FBS) and incubated at room temperature for at least 15 min. Afterwards, cells were washed with 1 ml wash buffer (TBS, 0.1% Triton X-100, 3% BSA, 2 mM $CaCl_2$, 0.1% $NaN_3$) and pelleted by centrifuging at $2300 \times g$ for 5 min at room temperature. The resultant supernatant was removed without disturbing the pellet. Cells were incubated with 100 μl wash buffer + 1:500 Monoclonal ANTI-FLAG-M1 antibody (Sigma, Cat. number F3040-1MG) for 30 min at room temperature whilst rotating. Next, 1 ml wash buffer was added, and the cells were pelleted at $2300 \times g$ for 5 min at room temperature. This step was repeated once more. Afterwards, cells were incubated with 100 μl wash buffer + 1:750 Alexa Fluor 647 Goat anti-mouse IgG (Abcam, Cat. number ab150115; 0.75 mg/ml in 50% glycerol), for 30 min whilst rotating in the dark. Pellets were washed twice by adding 1 ml wash buffer and pelleted at $2300 \times g$ for 5 min at room temperature. Cells were resuspended in 400 μl TBS, 2 mM $CaCl_2$, 0.1% $NaN_3$ and analysed by flow cytometry.

## Genetic complementation of AGR2

To genetically complement IRE1β/α expressing dual UPR reporter CHO cells with low levels of AGR2, retroviral transduction was performed (IRE1β/α; AGR$^+$ cells, Fig. EV5B). Cells were targeted with retrovirus expressing FLAG-M1-tagged AGR2 and puromycin selection marker. HEK293T cells were split onto 6-cm dishes 24 h prior to co-transfection of pBABE-pure plasmid encoding FLAG-M1 AGR2 (UK2881) with VSV-G retroviral packaging vectors (UK40-42), using TransIT-293 Transfection Reagent (Mirus). Sixteen hours after transfection, medium was changed to medium supplemented with 1% (w/v) BSA (Sigma). Retroviral infections were performed following a 24-h incubation by diluting 0.45-μm filter-sterilised cell culture supernatants at a 1:1 ratio into CHO cell medium supplemented with 10 μg/ml polybrene (8-ml final volume) and adding this preparation to IRE1β/α expressing CHO cells ($1 \times 10^6$ cells seeded onto 10-cm dishes 24 h prior to infection).

Infections proceeded for 8 h, after which viral supernatant was replaced with fresh medium. Forty-eight hours later, the cells were split into four 10-cm dishes. Five days after transfection, single cells were treated with puromycin. Serial dilution allowed to extract single clones whose phenotype was analysed by flow cytometry.

To genetically complement IRE1β/α expressing dual UPR reporter CHO cells with high levels of AGR2, cells were transfected with linearised plasmid DNA (IRE1β/α; AGR$^{++}$ cells, Fig. EV5B). In total, 4 μg of a plasmid encoding FLAG-M1-tagged AGR2 (preceded by a strong EF1 promoter) and puromycin resistance marker (UK2718) was used for transfection of a 50% confluent 10-cm dish. After 72 h, cells were treated with 8 mg/ml puromycin. Serial dilution allowed to extract single clones whose phenotype was analysed by flow cytometry. Puromycin was stopped 3 weeks after transfection.

## Mammalian cell lysis

Cell lysis was performed as described previously (Amin-Wetzel et al, 2017). In brief, adherent cells were grown in 10-cm dishes and treated as described above. The dishes were then transferred to ice and cells were washed in PBS and harvested in PBS + 1 mM EDTA with a cell scraper. The collected cells were spun at $370 \times g$ for 5 min at 4 °C. Cells were lysed in lysis buffer (1% Triton X-100, 150 mM NaCl, 20 mM HEPES-KOH pH 7.5, 10% glycerol, 1 mM EDTA, 1 mM phenylmethylsulphonyl fluoride (PMSF), 4 mg/ml Aprotinin, and 2 g/ml Pepstatin A, 2 mM Leupeptin). For analysis of potential mixed disulphides (Fig. 3C) the lysis buffer was further supplemented with 20 mM N-Ethylmaleimide (NEM). After 15 min of lysis on ice, cells were spun at $21,130 \times g$ for 10 min at 4 °C. The supernatant was transferred to a fresh tube and, when necessary, protein concentration measured with Bio-Rad protein assay (Bio-Rad).

## Immunoprecipitation (IP)

To analyse IRE1 variants expressed from the endogenous *ERN1* locus by MS (Fig. 1D), Protein A sepharose 4B beads (Zymed Invitrogen) were equilibrated in lysis buffer (see mammalian cell lysis). To reduce non-specific background binding, lysates were pre-cleared by adding 20 μl lysis buffer equilibrated beads and 1 μl rabbit non-immune serum per sample. After incubation for 1 h at 4 °C, samples were spun at $850 \times g$ at 4 °C and the supernatant was transferred into a new tube. Next, 20 μl lysis buffer equilibrated beads and 1 μl anti-IRE1α cytosolic domain (NY200) (Bertolotti et al, 2000) per sample were added to lysates and left rotating for 16 h at 4 °C. The beads were then washed three times with 1 ml lysis buffer (and spun down at $850 \times g$ at 4 °C), and residual liquid was removed using a syringe after the final washing step. The protein was eluted from the beads in SDS sample buffer containing 20 mM DTT.

For IP of FLAG-M1-tagged AGR2 (Fig. 3C), the lysis buffer was complemented with 10 mM $CaCl_2$ and 20 mM N-ethylmaleimide (NEM). Equal volumes of the cleared and normalised lysates were incubated with 20 μl of anti-FLAG-M1 affinity gel (Sigma) for 60 min at 4 °C, rotating. The beads were then recovered by centrifugation for 1 min at $5000 \times g$ and washed three times with TBS/Ca buffer (50 mM Tris, pH 7.4, with 0.15 M NaCl and 10 mM $CaCl_2$). The proteins were eluted in 35 μl of 2× SDS sample buffer

(without DTT) for 10 min at 70 °C. The beads were then sedimented and the supernatants were transferred to new tubes to which 50 mM DTT was added 'reduced' or an equal amount of water ('non-reduced'). Equal sample volumes were analysed by SDS-PAGE and immunoblotting as described below.

## Reducing/non-reducing SDS-PAGE and immunoblotting

After separation by SDS-PAGE on standard polyacrylamide Tris-glycine gels, the proteins were transferred onto PVDF membranes (pore size 0.45 μm, Sigma). The membranes were blocked with 5% (w/v) dried skimmed milk in TBS (25 mM Tris-HCl pH 7.5, 150 mM NaCl) and incubated with primary antibodies followed by IRDye fluorescently labelled secondary antibodies (LI-COR). The membranes were scanned with an Odyssey near-infrared imager (LI-COR). Primary antibodies and antisera against human IRE1α LD (Shemorry et al, 2019), mouse IRE1α serum (NY200) (Bertolotti et al, 2000), hamster BiP (chicken anti-BiP (Avezov et al, 2013)), eIF2α (mouse anti-eIF2α (Scorsone et al, 1987)) and monoclonal anti-FLAG-M1 (Sigma) were used.

Coomassie-staining was carried out with Instant Blue (Expedeon). Signal quantitation from SDS-PAGE gels or from immunoblots was carried out using the ImageJ software (NIH). For quantitative immunoblotting (Fig. EV5B), a precast gel NuPAGE™ 4 to 12%, Bis-Tris, 1.0–1.5 mm, Mini Protein Gel (Thermo Fisher) was used.

## Mass spectrometry

To validate IRE1 variants expressed from the endogenous *ERN1* locus of CHO cells (Fig. 1E), lysates were subjected to IP (as described above) using an anti-IRE1α cytosolic domain (NY200) antibody (Bertolotti et al, 2000). After the last washing step, samples were digested using the iST sample preparation kit (Preomics) following the manufacturer's recommendations.

LC-MSMS data was acquired on an Orbitrap Fusion Lumos coupled to an RSLC3000 via an EASYspray source using a 50 cm PepMap RSLC C18 EASYspray column. The UPLC was operated with solvent A (0.1% formic acid) and solvent B (80% acetonitrile, 0.1% formic acid) with peptides fractionated using a gradient rising from 7 to 37% solvent B by 58 min and 95% B by 62 min. Source voltage was maintained at 1.5 kV with data acquired from *m/z* 350 to 1500 in the Orbitrap at 120,000 FWHM. Peptides were fragmented using HCD activation at 34% collision energy, and MSMS spectra were generated in the ion trap using 1.0e4 AGC target, a maximum injection time of 250 ms and a cycle time of 2 s. Data was processed in Maxquant 2.1.0.0 using a Uniprot Chinese hamster database (downloaded 19/08/21). Carbamidomethyl (C) was set as a fixed modification and oxidation (M) and acetyl (protein N-terminus) set as variable modifications with LFQ and iBAQ enabled.

## Protein purification

### Human IRE1β luminal domain variants
MBP-IRE1β LD-His$_6$ variants (UK2796, UK3028, UK2986, UK3053, UK3155) were encoded on a pET-derived vector (Novagen) as fusion proteins and expressed in T7 Express lysY/Iq *E. coli* cells (NEB).

The following protocol refers to quantities used for processing 6 l bacterial expression culture. Bacterial cultures were grown at 37 °C in LB medium containing 100 mg/ml ampicillin until an OD$_{600\,nm}$ of 0.6–0.8 was reached. Expression was induced with 0.5 mM IPTG and the cells were incubated for 16 h at 18 °C. After sedimentation of the cells by centrifugation, the pellets were resuspended in TNGM buffer (50 mM Tris-HCl pH 7.4, 500 mM NaCl, 10% glycerol, 1 mM MgCl$_2$). The cell suspension was supplemented with 0.1 mg/ml DNaseI and protease inhibitors (2 mM PMSF, 4 mg/ml Pepstatin, 4 mg/ml Leupeptin, 8 mg/ml Aprotinin) and lysed by repeated passage through a high-pressure homogenizer (EmulsiFlex-C3, Avestin). After clarification of the lysates by centrifugation at 45,000×*g* for 30 min the purifications were performed in two steps: Ni-NTA followed by MBP affinity purification. The supernatant was removed and incubated for 60 min at 4 °C with 3 ml Ni-NTA agarose (bed volume, Qiagen). The matrix was transferred to a gravity-flow column and washed two times with 50 ml of TNGM supplemented with 30 mM imidazole. Next, the beads were washed with 50 ml wash buffer (50 mM HEPES-KOH pH 7.4, 300 mM NaCl, 5% glycerol). The flow-through was collected after a wash with one-bed volume of elution buffer (50 mM HEPES-KOH pH 7.4, 300 mM NaCl, 5% glycerol, 250 mM imidazole).

In total, 1.5-ml amylose resin (bed volume) was added to the elution and incubated overnight at 4 °C whilst rotating. The beads were washed with 50 ml HK buffer (50 mM HEPES-KOH pH 7.4, 150 mM KCl) and eluted (after a wash with one-bed volume of elution buffer) in HK supplemented with 10 mM maltose. The protein solutions were concentrated using 30 kDa MWCO centrifugal filters (Amicon Ultra; Merck Millipore). The sample was then separated on a Superdex 200 10/300 GL gel filtration column equilibrated in HK buffer and appropriate fractions collected and concentrated using 30 kDa MWCO centrifugal filters. A second gel filtration was performed to remove residual contaminating species on a HiScale S200 increase equilibrated in HK buffer. Appropriate fractions were collected and concentrated using 30 kDa MWCO centrifugal filters, flash-frozen, and stored at −80 °C.

For BLI probes UK2796 and UK3028, 10 μM biotin was added to the LB medium to enhance biotinylation of the AviTag. Note that for cysteine-containing proteins (UK2796, UK2986, UK3155) 1 mM TCEP was included in the buffer from the first Ni-NTA column wash onwards.

### Human IRE1α luminal domain variants
IRE1α LD variants (UK2246, UK2048) were encoded on a pET-derived vector (Novagen) as fusion proteins and expressed in T7 Express lysY/Iq *E. coli* cells (NEB).

Protein purification was performed as described in (Kono et al, 2017). Bacterial cultures were grown, induced, and lysed as described above. After clarification of the lysates by centrifugation at 45,000×*g* for 30 min the supernatant was removed and incubated for 60 min at 4 °C with Ni-NTA agarose (Qiagen) (0.5 ml per litre of bacterial culture). The matrix was washed two times with 50 ml of TNGM supplemented with 20 mM imidazole). The matrix was transferred to a gravity-flow column and the flow-through was collected after a wash with one-bed volume of elution buffer (50 mM Tris-HCl pH 7.4, 100 mM NaCl, 10% glycerol, 250 mM imidazole). The protein solutions were concentrated using 30 kDa

MWCO centrifugal filters (Amicon Ultra; Merck Millipore), flash-frozen and stored at −80 °C.

The purification of the fluorescently labelled IRE1α LD (UK2048) was performed as described above with 1 mM TCEP contained in all buffers. Eluted fractions were buffer exchanged into HKMT buffer (50 mM HEPES-KOH pH 7.4, 150 mM KCl, 10 mM MgCl₂, 1 mM TCEP) using a CentiPure P10 desalting column (Generon) and labelled with threefold molar excess of Oregon Green-iodoacetic acid (ThermoFisher) to make IRE1 LD R234C-OG (UK2048). The reaction proceeded at room temperature in the dark overnight and was quenched by the addition of 5 mM DTT. The reaction mixture was passed through a CentiPure P10 gravity-desalting column (Generon) equilibrated in HKM buffer and afterwards through a Superdex 200 10/300 GL gel filtration column equilibrated in HKG (50 mM HEPES-KOH pH 7.4, 150 mM KCl, 10% (v/v) glycerol) buffer. Appropriate fractions were collected, concentrated, flash-frozen and stored at −80 °C.

### Mouse AGR2 variants

His₆-SUMO-AGR2 variants (UK2794, UK2795, UK2816, UK2833, UK2835) were encoded on a pET-derived vector (Novagen) as fusion proteins and expressed in T7 Express lysY/Iq *E. coli* cells (NEB).

Bacterial cultures were grown, induced, and lysed followed by a Ni-NTA affinity purification as described above. After elution, 1.5 µg/ml His₆-SENP2 (UK2668) and 1 mM TCEP were added to the eluates and incubated overnight at 4 °C, whilst being dialysed against HK buffer. To remove the cleaved His₆-SUMO-tag and the His₆-SENP2 the solution was again incubated with Ni-NTA agarose for 60 min at 4 °C. After passing the sample through a gravity-flow column the final eluate was collected, concentrated using 30 kDa MWCO centrifugal filters, flash-frozen and stored at −80 °C.

### Analytical size-exclusion chromatography (SEC)

To assess the oligomeric state of wild-type GFP-tagged IRE1β LD (UK2986) and a cysteine-free version of it (UK3028) in the presence and absence of AGR2 variants, SEC was performed (Figs. 4A,B, EV2A,D and EV3A–E). Samples were run through a SEC-3 HPLC column (300 Å pore size; Agilent Technologies) on an Agilent Infinity HPLC system equilibrated in HK buffer at a flow rate of 0.3 ml/min. Samples were pre-incubated in a final volume of 20 µl for 30 min at 30 °C before clarification at 21,130×*g* for 5 min and subsequent injection of 10 µl. Runs were performed at 25 °C and $A_{280nm}$ absorbance and green fluorescence (excitation 488 nm and emission 507 nm) traces were recorded. For Fig. 4A, fractions were collected in 30 s time slices and subjected to analysis of their protein content after TCA precipitation.

### TCA precipitation

To analyse the protein content of fractions collected during SEC via SDS-PAGE (Fig. 4A), TCA precipitation was performed (Link and LaBaer, 2011). 0.2% Triton X-100 (v/v) and 0.11 volumes of ice-cold 100% TCA were added to each protein sample. Tubes were incubated on ice for 10 min followed by the addition of 500 µl ice-cold 10% TCA. After another incubation for 20 min on ice, samples were spun 20,000×*g* for 30 min. The supernatant was carefully removed and 500 µl acetone was added. After another centrifugation (20,000×*g* for 10 min) the supernatant was removed, and the

pellet dried for 30 min under a hood. The pellets were resuspended in alkaline SDS sample buffer (130 mM Tris-HCl pH 7.4, 10 mM EDTA pH 8, 3.3% SDS, 12% glycerol, 0.012 Bromphenol Blue), incubated at 72 °C for 10 min on loaded on an SDS-PAGE.

### Bio-Layer interferometry

All BLI experiments (Figs. 3A,B, 5C, and EV1C) were conducted on the FortéBio Octet RED96 System (Pall FortéBio) using an HK buffer supplemented with 0.05% Triton X-100 and 1 mM TCEP. Streptavidin (SA)-coated biosensors (Pall FortéBio) were hydrated in reaction buffer for 10 min prior to use. Experiments were conducted at 30 °C. BLI reactions were prepared in 200 µl volumes in 96-well microplates (greiner bio-one). Ligand loading was performed for 300–600 s at a shaking speed of 600 rpm until a binding signal of 2 nm was reached. The immobilised ligand sensor was then baselined in reaction buffer for at least 200 s.

Subsequent exposure to increasing concentrations of AGR2 was performed at 600 rpm shake speed with 700–900 s of association or dissociation. Data were processed in Prism 9 (GraphPad). Data were normalised to the signal after the first wash step. Plotting the change in signal (maximum of association subtracted by minimum of dissociation) for each AGR2 concentration allowed to disregard the small irreversible binding component at higher concentrations and extract a $K_D$ of binding. Note that IRE1 BLI probes were not subjected to in vitro biotinylation. Loading relied on endogenous biotinylation of a fraction of AviTagged- IRE1 LD proteins in *E. coli*.

### BRET

To assess the effect of AGR2 to IRE1β LD's monomer–dimer equilibrium a BRET-based assay was employed. A BRET donor IRE1β LD-nanoluc (UK3155) was incubated with an IRE1β LD-mGL (UK2986) or mGL (UK3060) acceptor for 30 min in a 384-well microplate (low volume, Corning) prior measurements with a CLARIOstar plate reader.

For steady-state readings (Figs. 6B and EV4A), 20 nM donor was incubated with 4 µM acceptor for 30 min followed by a luminescence spectral scan from 392 nm to 600 nm.

For kinetic readings (Fig. EV4B,D), indicated concentrations of AGR2 variants (UK2794, UK2795, UK2816, UK2833, UK2835) or unlabelled IREβ LD competitor (UK2796) were added to the well, and the reading started immediately. Signals were recorded for the donor and acceptor window (425–491 nm = donor signal and 480–528 nm = acceptor signal) every 20 s. To correct for the background signal due to the overlap of donor emission at the acceptor wavelength, a donor-only measurement was included to determine the corrected BRET:

$$Corrected\ BRET\ ratio = \frac{Acceptor\ signal_{Sample}}{Donor\ signal_{Sample}} - \frac{Acceptor\ signal_{Donor\ only}}{Donor\ signal_{Donor\ only}}$$

The resultant BRET ratios were plotted against time and analysed with the Prism 9 (GraphPad) software. To obtain reaction parameters, BRET ratios were normalised by setting their respective low BRET plateau at steady state to 1 (Fig. 6D). Fitting to a one-phase exponential decay allowed to obtain reaction parameters plotted in Fig. 6C,E. The former contains a plot of the difference in BRET (baseline minus AGR-induced plateau, ΔBRET) as a function

of AGR2 concentration fitted to a one-site binding function. The latter shows a plot of rate of BRET decrease as a function of AGR2 concentration fitted to a one-site binding function + background value. Note that the ΔBRET/Δt at [AGR2] = 0 is predicted to approach the off-rate of the IRE1β LD dimer determined by the addition of an unlabelled competitor (Fig. EV4E). However, the lower values of the observed 'Y' intercept of the plot in 6E likely reflects an error arising from inability to measure change in BRET as AGR2 concentrations approach zero.

## Mass photometry by interferometric scattering microscopy (ISCAT)

The mass photometry experiments (Fig. EV2C,E) were carried out in filtered HK reaction buffer on TwoMP instrument (Refeyn, UK) at room temperature, i.e., ~21 °C. Ready-to-use sample carrier slides (Refeyn) and sample well cassettes (six sample wells/cassette, Refeyn) were used to measure samples of 20 µl total volume. Samples were prepared 10x and incubated for 30 min at room temperature before diluting them 1:10 (18 µl reaction buffer + 2 µl sample) on the coverslip.

For calibration, standard protein solutions of Bovine serum albumin (BSA), Immunoglobulin G (IgG), and thyroglobulin protein (Tg) were used to generate the mass calibration of the contrast intensity to mass values. Mass photometry data was acquired with 10.9 µm × 4.3 µm instrument field of view and collected for 60 s at a 50 Hz frame rate on a 46.3 µm$^2$ detection area. At least $5 \times 10^3$ particles were detected in each acquisition. The resulting video data was analysed using DiscoverMP software provided by the instrument manufacturer (Refeyn, UK). Raw contrast values were converted to molecular mass using the standard mass calibration.

## Fluorescence detection system sedimentation velocity analytical ultracentrifugation (FDS-SV-AUC)

Samples of GFP-tagged IRE1b LD in 50 mM HEPES-KOH pH 7.4, 150 mM KCl, 4 mM TCEP, 0.05% (v/v) Tween-20, were centrifuged at 45,000 rpm at 20 °C in an An50Ti rotor using an Optima XL-I analytical ultracentrifuge (Beckmann) equipped with a fluorescence optical detection system (Aviv Biomedical) with fixed excitation at 488 nm and fluorescence detection at >505 nm. Data were processed and analysed using SEDFIT 16.36 (Schuck, 2013) according to the published protocol for high-affinity interactions detected by fluorescence (Chaturvedi et al, 2017). Data were plotted with Prism 9.5.1 (GraphPad) or GUSSI (Brautigam, 2015).

## RNA isolation and reverse transcription

Total RNA was purified from cells with the Invitrogen™ TRIzol™ Plus RNA Purification Kit. The RNA samples were treated with DNaseI for 1 h at 37 °C. This was followed by the addition of 50 mM EDTA and heat inactivation of the enzyme at 65 °C for 10 min. As a quality control, the RNA was analysed with the Agilent Bioanalyzer RNA 6000 pico assay. All RNA samples used for further processing had an RNA integrity number (RIN) of 7.3 or higher. Reverse transcription was carried out using the Thermo Scientific™ RevertAid First Strand cDNA Synthesis Kit. The cDNA was diluted up to 1:4.

## PCR analysis of *XBP1* mRNA splicing

Fragments of XBP1S and XBP1U were amplified from cDNA by PCR with NEB Q5® High-Fidelity 2X Master Mix and primers flanking the splice site recognised by IRE1 (Table 2) using the following PCR conditions: 94 °C for 4 min, 94 °C for 20 s, 65 °C for 10 s, 72 °C for 45 s and the last three steps repeated 35 times. The 255 bp fragment of XBP1U and the 229 bp fragment of XBP1S were separated in a 3% agarose gel by electrophoresis and stained with SYBR Green nucleic acid gel stain. In addition, a hybrid band, migrating as a fragment of approximately 280 bp, was observed. The percentage of XBP1S was quantified by determining the band intensity with Fiji, v1.53c and analysing the data with OriginPro, version 2023 (OriginLab Corporation, Northampton, MA, USA).

## qPCR

qPCR of RIDD targets was performed with the Bio-Rad CFX384 Touch Real-Time PCR Detection System using the Applied Biosystems™ PowerUp™ SYBR™ Green Master Mix and specific primers for each target (Table 2). Every sample was measured in triplicate and the signal of mock samples that did not undergo reverse transcription was measured to exclude cDNA or gDNA contamination. The mRNA expression was determined relative to 4µ8c treated cells and normalised to the signal of *Rpl27* and *GAPDH* as described by (Taylor et al, 2019).

# Data availability

This study includes no data deposited in external repositories.

# Peer review information

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

## Acknowledgements

We thank the Huntington lab for access to the Octet machine, the CIMR flow cytometry core facility team (Reiner Schulte, Chiara Cossetti and Gabriela Grondys-Kotarba) and the CIMR proteomics facility (Robin Antrobus, Harriet Parsons, John Suberu). We also thank Deborah Fass (Weizmann Institute) and Gunnar Hannsson (University of Gothenberg) for sharing mammalian expression vectors of human MUC2 fragments, and Avi Ashkenazi (Genentech) for the monoclonal antibody against human IRE1α LD. This work was supported by Wellcome Trust Principal Research Fellowship to DR (Wellcome 224407/Z/21Z), Medical Research Council DTP and Gates Cambridge PhD programme funding (MRC 2304568) to LN, MRC Programme code (MC_U105184326) to SHM, Fonds voor Wetenschappelijk onderzoek (FWO) Vlaanderen: 1228923N to EC, G017521N to SJ, ERC consolidator grant: DCRIDDLE- 819314 to SJ and Erasmus+ for CW and NF.

## Author contributions

**Lisa Neidhardt**: Conceptualisation; Data curation; Formal analysis; Supervision; Validation; Visualisation; Writing—original draft. **Eva Cloots**: Investigation. **Natalie Friemel**: Data curation; Formal analysis; Validation. **Caroline A M Weiss**: Data curation; Formal analysis; Supervision. **Heather P Harding**: Data curation; Supervision; Validation; Investigation; Methodology; Project administration. **Stephen H McLaughlin**: Data curation; Formal analysis. **Sophie Janssens**: Investigation. **David Ron**: Conceptualisation; Supervision; Funding acquisition; Investigation; Project administration; Writing—review and editing.

## Disclosure and competing interests statement

The authors declare no competing interests.

# Expanded View Figures

**Figure EV1.  IRE1β luminal domain (LD) cysteines are dispensable to its interaction with AGR2.**

(A) Two-dimensional contour plot of CHOP::GFP and XBP1s::Turquoise signals from dual UPR reporter CHO cells expressing an IRE1β/α chimera wild-type (wt) or mutant version lacking the two cysteines of the IRE1β LD [LD ΔC Clone (Cl.) 1] from the endogenous *ERN1* locus. Where indicated cells were treated with the ER stressors tunicamycin (Tm) or thapsigargin (Tg). For reference, the signal of the IRE1α ΔLD parental cells is indicated in grey. Representative plot of three independent experiments is shown. (B) Upper panel: Histograms of CHOP::GFP and XBP1s::Turquoise signal from cells described in (A) transiently transfected with FLAG-M1-AGR2. Lower panel: Bar diagram of the mean ± SD of the indicated signal from three independent repetitions of the experiment shown in the upper panel. Statistical analysis was performed by two-sided unpaired Welch's t test and significance is indicated by asterisks (**$P < 0.01$, ***$P < 0.001$, ****$<0.0001$). (C) Bio-Layer Interferometry (BLI)-derived signal from an IRE1β LD ΔC probe as a function of concentration of interacting AGR2 fitted to a one-site specific binding function with the indicated confidence interval (CI). Shown are the mean ± SD of three independent repetitions.

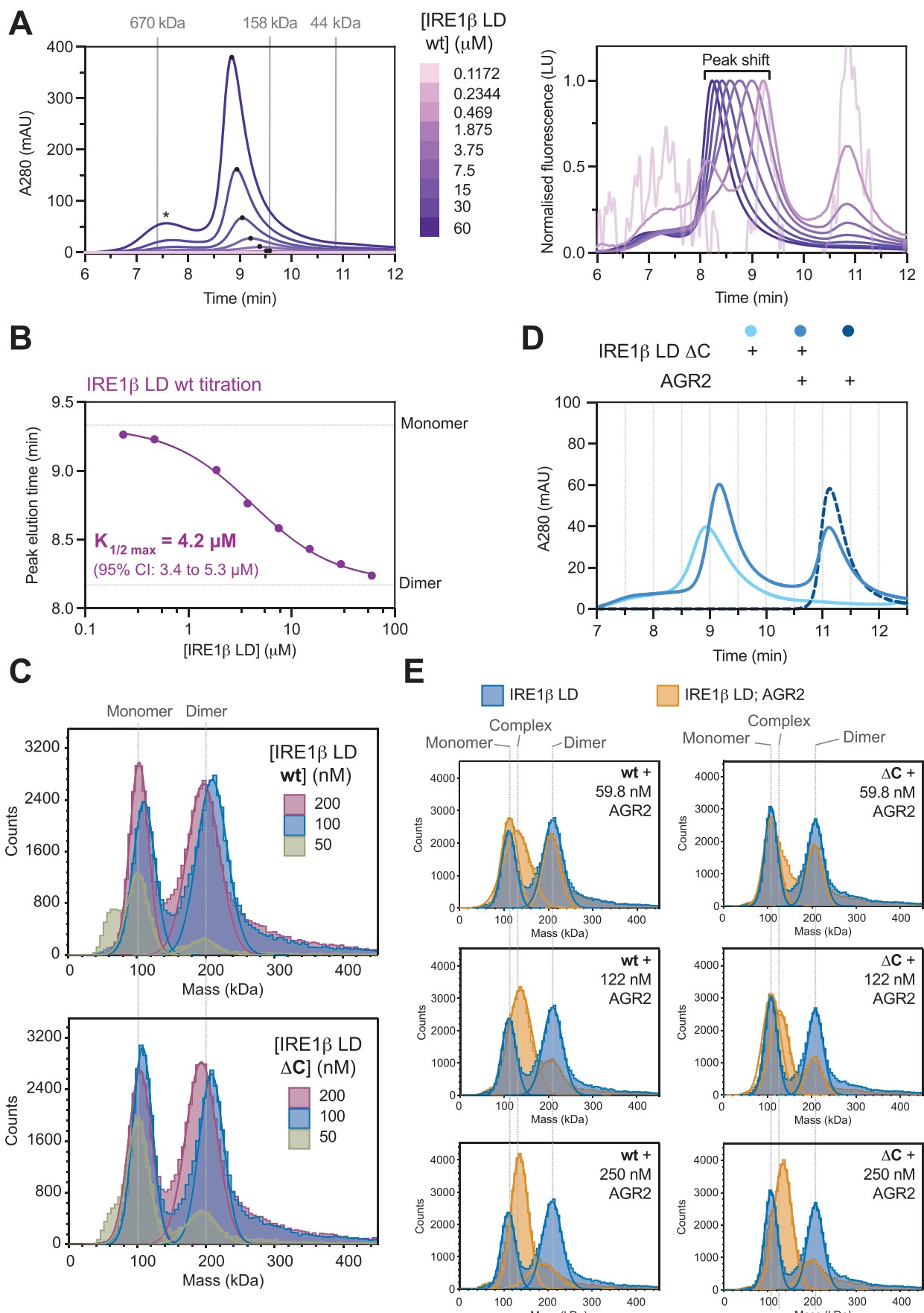

◀  **Figure EV2.  Both, wild-type (wt) IRE1β luminal domain (LD) and a cysteine-lacking mutant (△C), exist in a monomer–dimer equilibrium which is shifted towards the monomeric species in presence of AGR2.**

(**A**) Left panel: Size-exclusion chromatography (SEC) elution profiles of IRE1β LD samples at the indicated concentrations. The asterisk (*) marks a high molecular weight species that is a purification artefact as it was not in equilibrium with the main peak. Right panel: Normalised fluorescence trace of the same samples, plotted against peak elution signal. (**B**) Plot of the relation of peak elution time and protein concentration of IRE1β LD from (**A**) fitted to a sigmoidal function with the indicated confidence interval (CI). (**C**) Histograms of mass photometric analysis of IRE1β LD wt or LD △C at the indicated concentrations. Note that the molecular mass of the monomer is 116 kDa, since IRE1β LD variants were expressed as fusion proteins with maltose binding protein and GFP. (**D**) SEC protein absorbance as in (**A**) of GFP-tagged IRE1β LD lacking its cysteines (△C) with and without AGR2. Representative plot of three independent experiments is shown. (**E**) Histograms as in (**C**) of 100 nM IRE1β LD wt or △C in presence and absence of the indicated concentrations of AGR2.

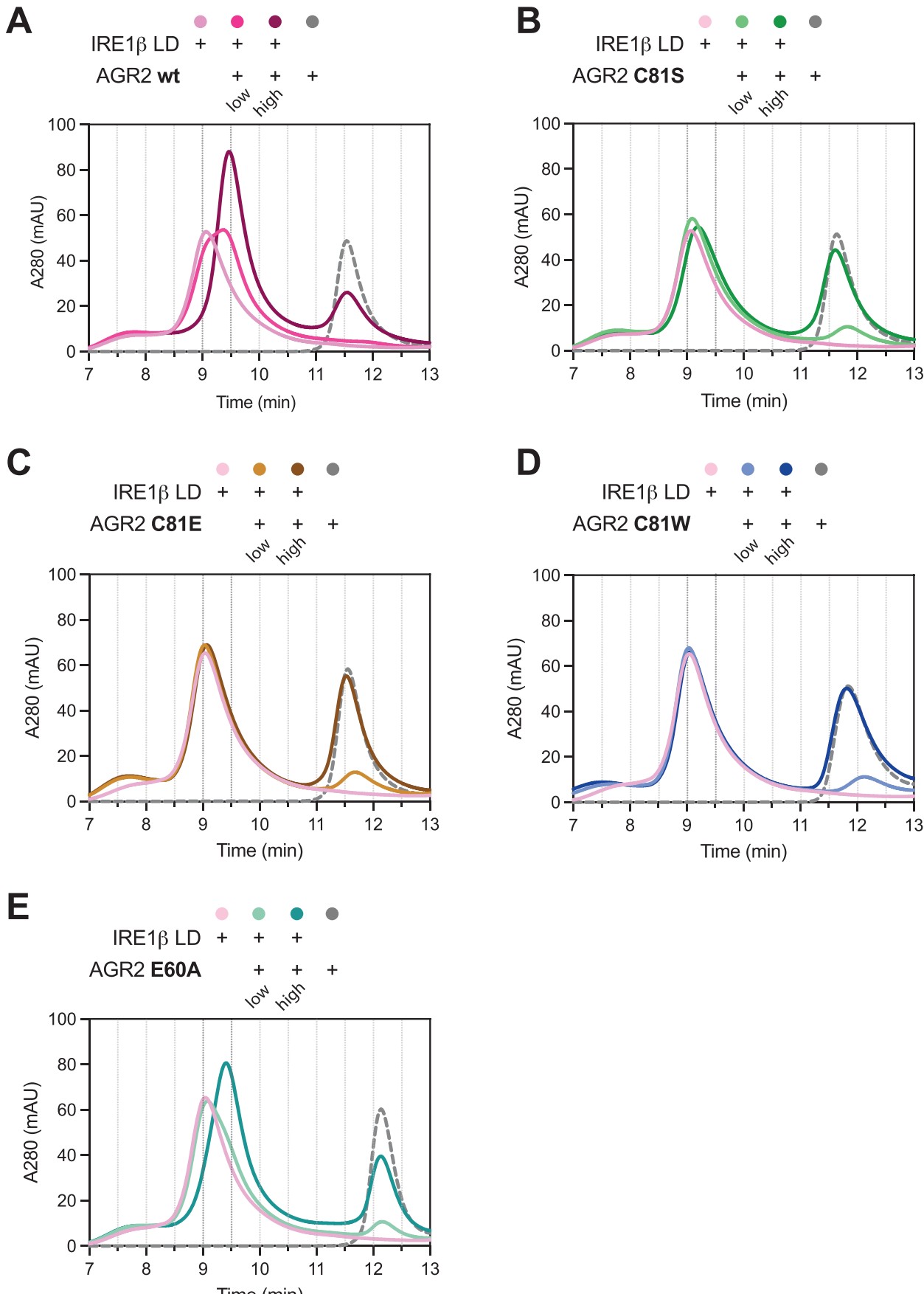

◄ **Figure EV3. Active-site mutations impair AGR2's effect on the oligomeric state of IRE1β's luminal domain (LD) in vitro.**

(**A–E**) Size-exclusion chromatography (SEC) elution profiles of IRE1β LD (15 μM) incubated with 10 μM (low) or 50 μM (high) of wild-type (wt) or mutant AGR2. AGR2 E60A is a monomeric mutant previously described in (Patel et al, 2013).

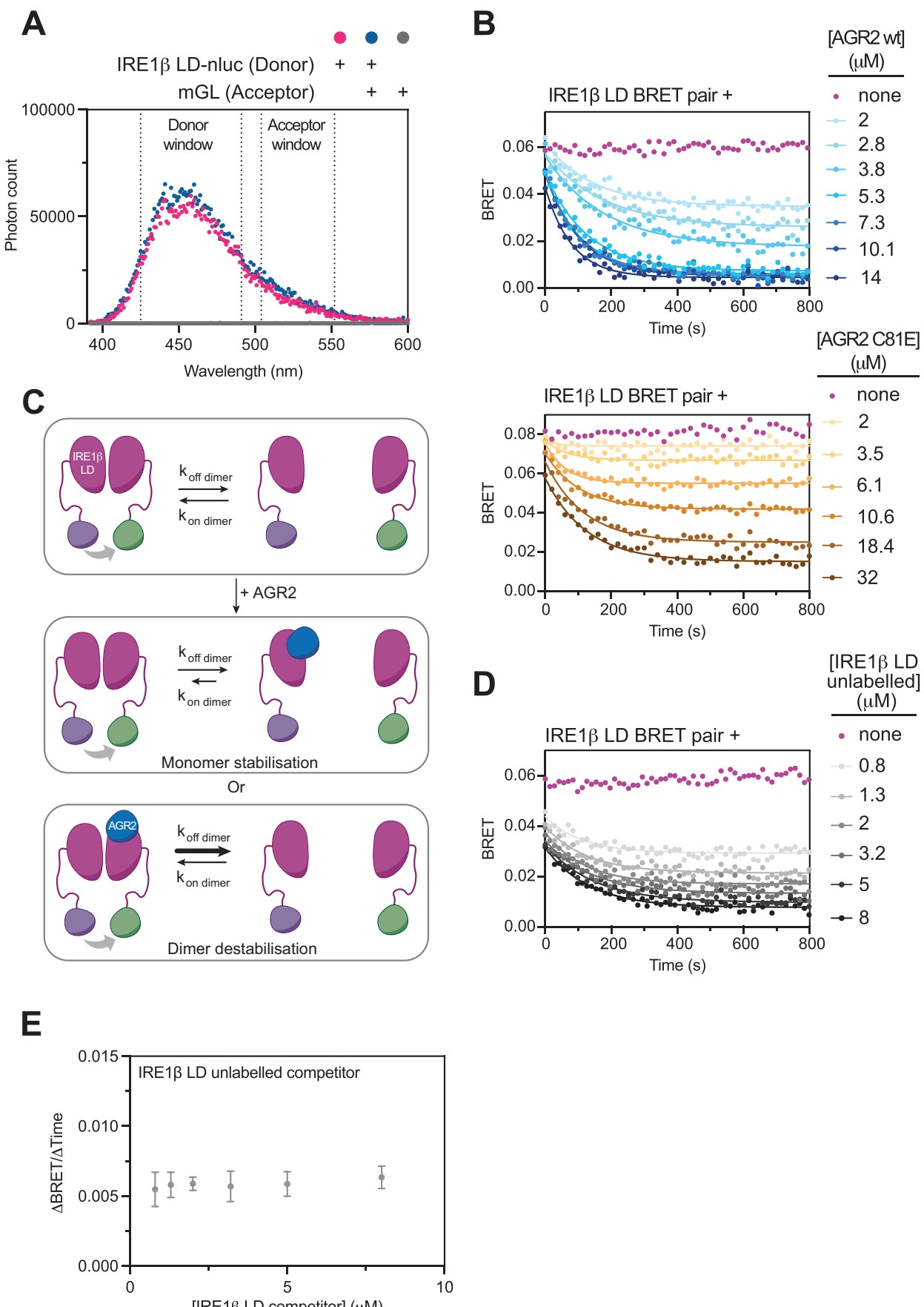

◀ **Figure EV4. Kinetics of IRE1β luminal domain (LD) dimerisation and AGR2's effect on it probed by bioluminescence resonance energy transfer (BRET).**

(A) Spectral scan as in Fig. 6B, with free monomeric Green Lantern (mGL) as an acceptor. (B) Plot of time-dependent change in IRE1β LD dimerisation-induced BRET following introduction of wild-type (wt) AGR2 and AGR2 C81E mutant, respectively. Traces were fitted to a one-phase exponential decay and parameters used to obtain reaction variables plotted in Fig. 6C, E. (C) Schematic depiction of IRE1β LD's monomer–dimer equilibrium. Two hypothetical components to AGR2 action are presented: AGR2 can bind to monomeric IRE1β LD and prevent re-dimerisation, resulting in decreased dimerisation on-rates. Whereas active destabilisation of dimeric IRE1β LD upon AGR2 binding would increase the 'off-rate' of the dimer. (D) As in (B) but following introduction of an unlabelled IRE1β LD competitor. (E) Plot of the rate of BRET decrease as a function of concentration of unlabelled IRE1β LD competitor. Shown are the mean ± SD of three independent repetitions.

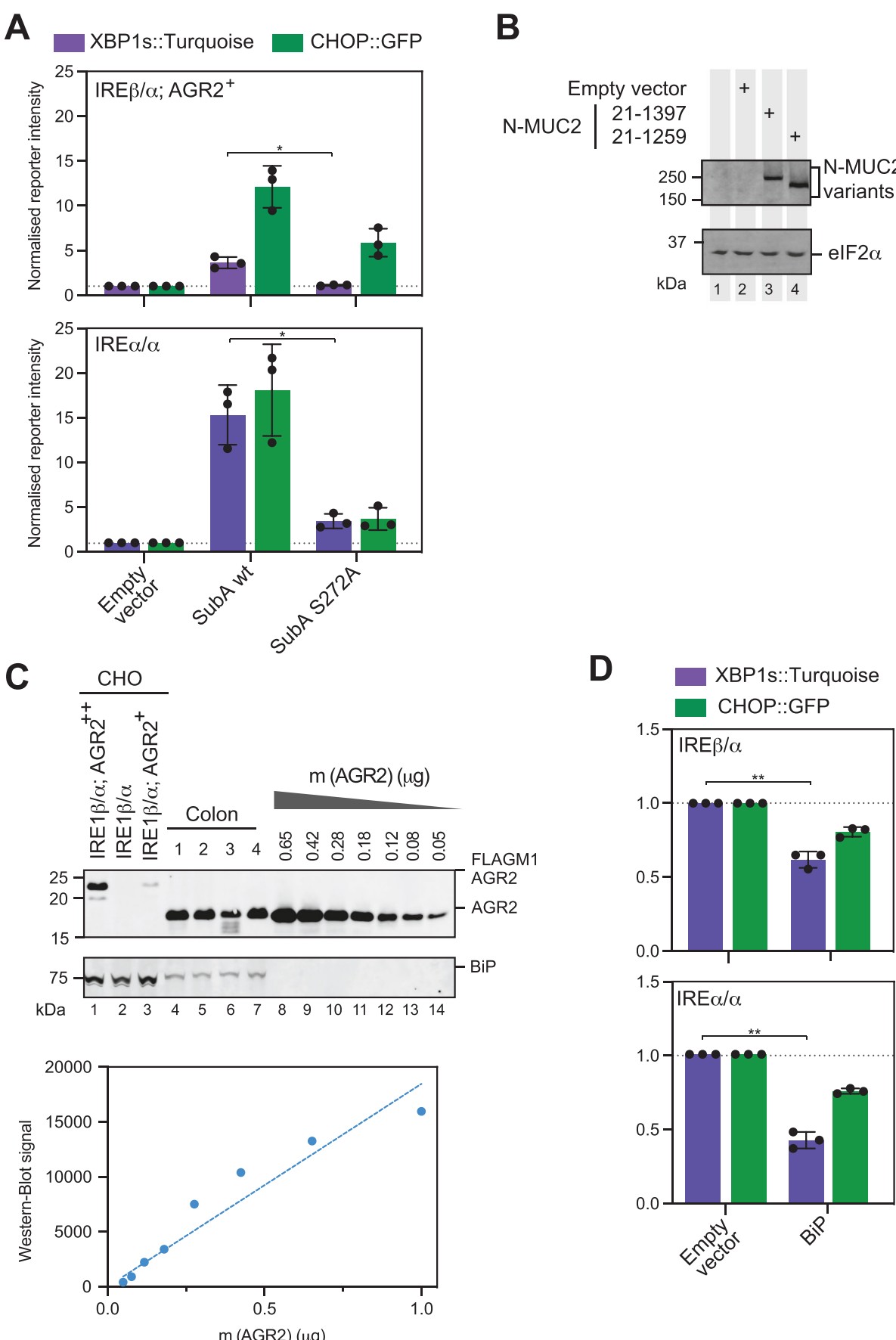

**Figure EV5. Expression of MUC2 fragments and AGR2.**

(A) Quantification of reporter signals from IRE1β/α; AGR2$^+$ and IRE1α/α expressing dual UPR reporter cells transiently transfected with SubA variants (mCherry as expression marker). The inactive SubA S272A mutant was used as control. Shown are the mean ± SD of three independent repetitions. Statistical analysis was performed by two-sided unpaired Welch's *t* test and significance is indicated by asterisks (*$P > 0.1$). (B) Representative immunoblot of IRE1β/α; AGR2 cells transiently transfected with the indicated MUC2 variants (from Fig. 7C, D). (C) Upper panel shows quantitative immunoblotting (using known quantities of purified bacterially expressed AGR2 to calibrate the assay) of AGR2 the indicated CHO cell lines and mouse colonic lysates. Lower panel shows calibration curve derived from known quantities of purified AGR2. Data points were fitted to a linear function. Assuming that 50% of the mass recovered from the colonic mucosa were cells, from which 10% are goblet cells (Kim and Ho, 2010), and that the ER constitutes 10% of cell volume total volume (Alberts, 2002) the concentration of AGR2 in the goblet cell ER is about 460 μM. IRE1β/α; AGR2$^+$ and AGR2 high expressing IRE1β/α; AGR2$^{++}$ cells had a concentration of 12 μM and 140 μM of AGR2 in their ER, respectively. (D) Quantification of reporter signals from IRE1β/α and IRE1α/α expressing dual UPR reporter CHO cells transiently transfected with BiP (mCherry as expression marker). Shown are the mean ± SD of three independent repetitions. Statistical analysis was performed by two-sided unpaired Welch's *t* test and significance is indicated by asterisks (**$P < 0.01$).

