## [Peer Review File · The EMBO Journal]

The IRE1 β -mediated unfolded protein response is repressed by the chaperone AGR2 in mucin producing cells

Lisa Neidhardt, Eva Cloots, Natalie Friemel, Caroline Weiss, Heather Harding, Stephen McLaughlin, Sophie Janssens, and David Ron

DOI: 10.15252/emboj.2023114737

Corresponding author(s): David Ron (dr360@medschl.cam.ac.uk) , David Ron (dr360@medschl.cam.ac.uk), Lisa Neidhardt (ln327@cam.ac.uk)

Review Timeline:

Submission Date:	14th Jun 23
Editorial Decision:	1st Aug 23
Revision Received:	16th Oct 23
Editorial Decision:	3rd Nov 23
Revision Received:	7th Nov 23
Accepted:	15th Nov 23

Editor: William Teale

Transaction Report:

Dear Lisa and David,

Thank you again for the submission of your manuscript entitled "Repression of the mucin producing cell-specific IRE1 β isoform by a mucin chaperone". We have now received reports from all three referees, which I copy below.

As you can see from their comments, all referees were supportive of publication in The EMBO Journal. Referees 1 and 2 suggest some strengthening experiments and controls; please consider these suggestions and respond in a document that addresses all points raised.

Based on the overall interest expressed in the reports, therefore, I would like to invite you to address the comments of all referees in a revised version of the manuscript. I should add that it is The EMBO Journal policy to allow only a single major round of revision and that it is therefore important to resolve the main concerns at this stage. I believe the concerns of the referees are reasonable and addressable, but please contact me if you have any questions, need further input on the referee comments or if you anticipate any problems in addressing any of their points. I am available to Zoom call if you would like to discuss referee 1 and referee 2's reports. Please, follow the instructions below when preparing your manuscript for resubmission.

I would also like to point out that as a matter of policy, competing manuscripts published during this period will not be taken into consideration in our assessment of the novelty presented by your study ("scooping" protection). Please contact me if you see a paper with related content published elsewhere to discuss the appropriate course of action.

Again, please contact me at any time during revision if you need any help or have further questions.

Thank you very much again for the opportunity to consider your work for publication. I look forward to your revision.

Best regards,

William

William Teale, Ph.D.
Editor
The EMBO Journal

When submitting your revised manuscript, please carefully review the instructions below and include the following items:

- 1) a .docx formatted version of the manuscript text (including legends for main figures, EV figures and tables). Please make sure that the changes are highlighted to be clearly visible.
- 2) individual production quality figure files as .eps, .tif, .jpg (one file per figure).
- 3) a .docx formatted letter INCLUDING the reviewers' reports and your detailed point-by-point response to their comments. As part of the EMBO Press transparent editorial process, the point-by-point response is part of the Review Process File (RPF), which will be published alongside your paper.
- 4) a complete author checklist, which you can download from our author guidelines ([https://wol-prod-cdn.literatumonline.com/pb-assets/embo-site/Author Checklist%20-%20EMBO%20J-1561436015657.xlsx](https://wol-prod-cdn.literatumonline.com/pb-assets/embo-site/Author%20Checklist%20-%20EMBO%20J-1561436015657.xlsx)). Please insert information in the checklist that is also reflected in the manuscript. The completed author checklist will also be part of the RPF.
- 5) Please note that all corresponding authors are required to supply an ORCID ID for their name upon submission of a revised manuscript.
- 6) We require a 'Data Availability' section after the Materials and Methods. Before submitting your revision, primary datasets produced in this study need to be deposited in an appropriate public database, and the accession numbers and database listed under 'Data Availability'. Please remember to provide a reviewer password if the datasets are not yet public (see <https://www.embopress.org/page/journal/14602075/authorguide#data deposition>). If no data deposition in external databases is needed for this paper, please then state in this section: This study includes no data deposited in external repositories. Note that

the Data Availability Section is restricted to new primary data that are part of this study.

Note - All links should resolve to a page where the data can be accessed.

8) For data quantification: please specify the name of the statistical test used to generate error bars and P values, the number (n) of independent experiments (specify technical or biological replicates) underlying each data point and the test used to calculate p-values in each figure legend. The figure legends should contain a basic description of n, P and the test applied. Graphs must include a description of the bars and the error bars (s.d., s.e.m.).

9) We would also encourage you to include the source data for figure panels that show essential data. Numerical data can be provided as individual .xls or .csv files (including a tab describing the data). For 'blots' or microscopy, uncropped images should be submitted (using a zip archive or a single pdf per main figure if multiple images need to be supplied for one panel). Additional information on source data and instruction on how to label the files are available at .

10) We replaced Supplementary Information with Expanded View (EV) Figures and Tables that are collapsible/expandable online (see examples in <https://www.embopress.org/doi/10.15252/embj.201695874>). A maximum of 5 EV Figures can be typeset. EV Figures should be cited as 'Figure EV1, Figure EV2" etc. in the text and their respective legends should be included in the main text after the legends of regular figures.

12) Our journal encourages inclusion of *data citations in the reference list* to directly cite datasets that were re-used and obtained from public databases. Data citations in the article text are distinct from normal bibliographical citations and should directly link to the database records from which the data can be accessed. In the main text, data citations are formatted as follows: "Data ref: Smith et al, 2001" or "Data ref: NCBI Sequence Read Archive PRJNA342805, 2017". In the Reference list, data citations must be labeled with "[DATASET]". A data reference must provide the database name, accession number/identifiers and a resolvable link to the landing page from which the data can be accessed at the end of the reference. Further instructions are available at .

At EMBO Press we ask authors to provide source data for the main manuscript figures. Our source data coordinator will contact you to discuss which figure panels we would need source data for and will also provide you with helpful tips on how to upload

and organize the files.

We realize that it is difficult to revise to a specific deadline. In the interest of protecting the conceptual advance provided by the work, we recommend a revision within 3 months (30th Oct 2023). Please discuss the revision progress ahead of this time with the editor if you require more time to complete the revisions. Use the link below to submit your revision:

Referee #1:

REVIEW

This manuscript by Neidhardt et al focuses is directed towards defining the mechanism of IRE1beta activation. The establish a novel chimeric system where the luminal domain of endogenous IRE1alpha in CHO cells is replaced with the luminal stress sensing domain of IRE1beta. They found that overexpression of the mucin chaperone AGR2 suppressed this constitutive activation of the IRE1alpha/beta chimera in cells and disrupted dimerization of the luminal domain in vitro. Further, they demonstrated that overexpression of the AGR2 substrate mucin could 'reactivate' IRE1alpha/beta chimeras, indicating that the interaction between the luminal domain of IRE1beta and AGR2 functions to sense specific stresses linked to mucin production.

This manuscript, especially in context of its accompanying manuscript by Cloots et al, provides significant new insight into the regulation of IRE1beta in specialized cells such as goblet cells. The experiments are well performed and convincing, especially with the combination of efforts across the two co-submitted papers. I have no specific issues with any of the experimental designs or conclusions, all of which seem well performed and analyzed. I only have a couple simple suggestions that would further highlight the specificity and importance of the model described in this manuscript, which are included below. However, as indicated above, this manuscript and the accompanying manuscript provide very nice complementary evidence that AGR2 functions as a repressor of IRE1beta activity through engagement with the IRE1beta luminal domain and represents a key factor to sense specific proteostasis stresses within goblet cells (e.g., mucin misfolding/misfolding).

COMMENTS.

1. The authors are well positioned to address the potential contributions of BiP to the regulation of IRE1beta using their reporter cells expressing IRE1alpha/beta. Including some experiments where BiP is overexpressed in these cells could reveal some potential cooperation between AGR2 and BiP in regulating IRE1beta activity through engaging the luminal domain. Either way, this result will not significantly impact the main conclusions of the study showing the importance of AGR2 in regulating IRE1beta activity, but they could demonstrate a more nuanced coordination between AGR2 and BiP in regulating IRE1beta activity in response to different types of stresses. This is alluded to in the text and seems like something that could be explored a bit more in this manuscript using the very nice tools established.
2. Along the same lines, it could be useful to overexpress a more canonical PDI (e.g., PDIA1 or PDIA6 both of which have been implicated in regulating IRE1alpha) in the reporter cells to demonstrate the specificity of IRE1alpha/beta repression for AGR2.
3. Inclusion of some immunoprecipitations of the IRE1alpha/beta-AGR2 interactions in cells, especially as it relates to the mucin experiments, would help support the model described within the manuscript. Notably, one would predict that overexpression of mucin should deplete AGR2 from IRE1alpha/beta, which could be reflected by reduced interactions observed by IP. These experiments would provide additional evidence to support the idea that IRE1beta-AGR2 interactions is a specific stress sensor of

mucin folding load within the ER .

Referee #2:

This study demonstrates the mechanism of activation of IRE1beta in a context recapitulating molecules expressed in goblet cells. The authors show that the chaperone AGR2 binds to the IRE1 Ld domain, resulting in the inactivation of the protein. However, the presence of mucins relieves this interaction, leading to the engagement of the IRE1 cytosolic response. Importantly, the deficiency of AGR2 leads to the constitutive activation of IRE1beta but not IRE1alpha. On the other hand, the induction of misfolded proteins promotes IRE1alpha activation in the presence of AGR2 without allowing the activation of IRE1beta. Overall, this study and the accompanying paper support the model that IRE1s act as guardians of proteostasis in mammals rather than directly detecting specific ligands. They sense different insults and are released from inhibition by their specific sensor chaperones: AGR2 for IRE1beta and BiP for IRE1alpha. This is a fundamental and important discovery in the cell homeostasis maintenance field.

The strength of this manuscript relies on the biochemical approaches that nicely describe the molecular complex involved. The technical comments below are aimed at strengthening the central message of the paper. Some of the comments may have been addressed in the accompanying paper (cited in the introduction and in the discussion).

- 1) While overall, the study is convincing, the fact that the chimeric construct (IRE1beta luminal domain and IRE1alpha cytosolic region) is constitutively active, as reported in Figure 1, could also be caused by conformational perturbation of fusing two proteins and/or from two different species mouse and chinese hamster. The study of a cell line with the entire alpha locus replaced with the entire locus IRE1beta locus could provide excellent additional controls.
- 2) The study relies on a model line recapitulating proteins expressed in goblet cells. However, using a cellular model expressing IRE1beta and AGR2 could strengthen the physiological relevance of the study. In that model, it would be possible to delete AGR2 and confirm that no specific IRE1beta ligands are expressed under physiological expression.
- 3) I could not find the nature of the cells used for the transfections-IP in Figure 3. Are those CHO cells deficient for endogenous IRE1 alpha? Overexpression of chaperones has the tendency to generate artifacts due to the plasticity of the folds that can be expressed upon overexpression. If technically feasible, I suggest the authors confirm these experiments using the stable lines described in Figure 1 and the AGR2 ko described in the appendix as controls. Moreover, the parallel comparison of both BiP and AGR2 binding to both IRE1 alpha and IRE1 beta would provide excellent controls that could support the possible scenario: IRE1 alpha is BiP controlled and beta AGR2 controlled.
- 4) The KEN-mediated differences observed in Figure 8 are difficult to interpret. These differences in XBP1s signaling could be due to slight structural imperfections caused by the chimera in the construct with the beta intracellular portion. Because the LD and transmembrane region of alpha is conserved in both cases, it is difficult to exclude that transmission of the signal is more effective when the alpha region LD is fused to the alpha KEN region than when the alpha region LD is fused to the beta KEN region. A possible additional experiment that would support the finding would be to repeat the experiment using the beta LD domain and monitoring constitutive activation, for example. If the beta/beta construct is less active than the beta/alpha construct, it could support the idea that the nature of the response is different.

Referee #3:

The unfolded protein response (UPR) is a homeostatic response of eukaryotic cells to cope with protein folding stress in the endoplasmic reticulum (ER). The central regulator of one of the three UPR branches is Ire1, a kinase and endonuclease that activates the transcription factor XBP-1 and cleaves a number of other mRNAs. Essential for Ire1 activation is the monomer-to-oligomer transition. How the monomer-oligomer equilibrium of Ire1 is regulated by accumulating misfolded proteins in the ER is debated since many years. One model proposes that misfolded proteins bind directly to Ire1 and promote Ire1 oligomerization thereby acting as positive regulator. A second model poses that Ire1 spontaneously dimerizes and that the Hsp70 chaperone of the ER, BiP, actively dissociate Ire1 dimers/oligomers thereby acting as negative regulator. Since misfolded proteins bind to BiP and thus titrate BiP away from Ire1, according to the second model, deconvolution of direct and indirect effects of misfolded proteins is far from trivial.

In mammalian cells there are two isoforms of Ire1, Ire1 α the major isoform expressed in most cells and induced by general protein misfolding in the ER and Ire1 β the isoform that is only expressed in some mucus producing cells, like goblet cells of the intestine, and induced by folding defects of mucins. To deconvolute the action of the Ire1 inducer, the authors exchanged the luminal domain of Ire1 α with the luminal domain of Ire1 β creating the Ire1 β/α chimera in CHO cells where full-length Ire1 β is not present. The chimeric Ire1 β/α protein was constitutively active as demonstrated by several orthologous assays. Coexpression of the mucin specific chaperone AGR2 repressed the activity and this repressive action is counteracted by the coexpression of the AGR2 substrate MUC2. The authors show by biolayer interferometry that AGR2 binds to the luminal domain of Ire1 β but not

Ire1 α . Using several orthologous in vitro assays with purified proteins including size exclusion chromatography, sedimentation velocity analytical ultracentrifugation and mass photometry, the authors demonstrate that the luminal domain of Ire1 β is in monomer-dimer equilibrium and that AGR2 shifts the equilibrium to the monomeric state. Using a bioluminescence resonance energy transfer (BRET) assay, the authors demonstrate that AGR2 directly binds to Ire1 β luminal domain dimers and induce monomerization. This seems to be an active process as monomerization rates in the presence of AGR2 are higher than in the presence of an excess of unlabeled Ire1 β protein and increases with increasing AGR2 concentrations. The authors generate a number of mutants of AGR2 that have reduced ability to monomerize Ire1 β in vitro and a diminished repressive effect on Ire1 β / α activity in CHO cells.

This study demonstrates beyond any reasonable doubt that (1) Ire1 β is spontaneously dimerizing/oligomerizing in vivo and constitutively active in the absence of its negative regulator, the molecular chaperone AGR2; (2) Ire1 β dimers are monomerized by AGR2; (3) amino acid replacements that reduce or prevent binding of AGR2 to Ire1 β inactivate the monomerization activity of AGR2 in vitro and its repression of Ire1 β activity in vivo.

In my opinion this is an outstanding study, experiments are thoroughly designed and meticulously performed, interpretation of the data is cautious and rigorous, and conclusions seem fully warranted. There are only a few minor comments.

Minor comments:

In expanded view Fig. EV3 the authors present elution profiles of size exclusion chromatography for IRE1 β in the absence of any interacting proteins or in the presence of different AGR2 wild type and mutant proteins. Normally the elution of column chromatography is given in volume units not time. It is curious that in panels A and B the time scale is 7 to 13 minutes whereas in panels C-E it is 7 to 13 seconds. The latter seems to be a typographic error.

Fig. 6E the authors present the increase in IRE1 β dissociation rates with increasing AGR2 concentrations and fit the data to a one site binding equation with the dissociation rate of IRE1 β n dimers in the absence of AGR2 as background. They give the $K_{1/2}$ max in units of $\mu\text{M}\cdot\text{s}^{-1}$. This is not correct. The resulting $K_{1/2}$ max will be in units of μM , because in the denominator of the equation for the one site binding reaction there is only $K_D + [\text{Ligand}]$ both of which should have the same units (mol/l).

Cover_Letter V3

16 October 2023

Re: Manuscript EMBOJ-2023-114737 - Decision

Dear Editor,

Thank you for your letter of 01-Aug-2023 detailing the reviewer's critiques of our paper. Thanks too to the reviewers for their time and effort. We are pleased by your invitation to submit a revised version addressing these criticisms. The following outlines a summary as well as a point-by-point list (with the reviewer's comments and *our response*) to the critiques articulated in your letter. Changes made in the manuscript are **marked in green** in the revised manuscript .docx formatted version.

Summary

The reviewers accepted the key conclusion of the study that the mucin chaperone AGR2 is a physiologically significant and selective repressor for the beta isoform of IRE1. They suggested experiments to further probe the specificity of the repressive action of AGR2 on IRE1 β signalling by comparing it to a broadly-expressed reference protein of the same PDI family. Following this proposal, we transiently transfected PDIA1, previously implicated in IRE1 α regulation (Yu et al., 2020) into IRE1 β/α expressing dual UPR reporter cells. Compared to AGR2 co-expression, presence of PDIA1 failed to repress the deregulated IRE1 β/α chimera (**Fig 2D** of the updated manuscript). This experiment reinforces the existence of a special relationship between AGR2 and IRE1 β .

The reviewers also noted the potential to address the contribution of ER Hsp70 chaperone BiP to IRE1 β signalling by employing our reporter cells. In line with these suggestions, we introduced SubA, a protease that inactivates BiP by cleaving its interdomain linker (Paton et al., 2006) into the reporter cells. As noted previously SubA-mediated BiP depletion strongly induced both IRE1 α/α signalling and the PERK dependent CHOP::GFP reporter (Amin-Wetzel et al., 2019), IRE1 β/α was less responsive to this intervention (**Fig EV5A** of the updated manuscript). In a reciprocal experimental design, BiP overexpression attenuated both, active IRE1 β/α (in absence of AGR2) and IRE1 α/α (upon ER stress induction). Together this suggests that whilst AGR2 selectively serves as a repressor for the β isoform, BiP governs IRE1 α signalling and has the potential to contribute to IRE1 β repression. In line with this notion, Bertolotti et al. (Bertolotti et al., 2000) (Fig. 3E, therein) showed that BiP co-IPs with IRE1 β in a stress-dependent manner.

A circuit involving endogenous AGR2 and MUC2 has been implicated in regulating IRE1 β (in the accompanying manuscript by Cloots et al.) and has been reconstructed here by co-expression of MUC2 and AGR2 using the fluorescent reporters in cells. The reviewers suggested to complement these findings with co-immunoprecipitation (IP) experiments to test the prediction that abundance of an IRE1 β -AGR2 complex would correlate negatively with the mucin load in the ER. We accept the logic behind this suggestion but note that we are unable to detect an IRE1 β/α -AGR2 complex in cells expressing IRE1 β/α from the endogenous *ERN1* locus. This likely reflects the low expression levels of the IRE1 β/α protein compounded by high off rates of the complex. This state of affairs all but preclude performing the experiment suggested by the reviewers in a system with a relevant concentration regime of IRE1 β/α .

Elsewhere the reviewers suggested complementing the suite of cell lines employed in this study (IRE1 α/α , IRE1 β/α and IRE1 α/β) with cells expressing IRE1 β/β from the endogenous *ERN1* locus. Whilst we recognise the theoretical utility of this approach,

we note that creating such a cell line (via homologous recombination) is expected to take many months and therefore exceeds the time frame normally allowed for review. However, we have addressed the reviewer's suggestion editorially (see point-by-point response below).

Point-by-point response

Reviewer 1:

1. The authors are well positioned to address the potential contributions of BiP to the regulation of IRE1beta using their reporter cells expressing IRE1alpha/beta. Including some experiments where BiP is overexpressed in these cells could reveal some potential cooperation between AGR2 and BiP in regulating IRE1beta activity through engaging the luminal domain. Either way, this result will not significantly impact the main conclusions of the study showing the importance of AGR2 in regulating IRE1beta activity, but they could demonstrate a more nuanced coordination between AGR2 and BiP in regulating IRE1beta activity in response to different types of stresses. This is alluded to in the text and seems like something that could be explored a bit more in this manuscript using the very nice tools established.

*This point has been addressed experimentally. BiP levels were manipulated via two approaches: 1) Introduction of SubA, a protease that inactivates BiP by cleaving its interdomain linker (Paton et al. 2006) strongly activated IRE1 α/α signalling and the PERK dependent CHOP::GFP reporter. In contrast, IRE1 β/α appeared less subordinate to SubA-mediated BiP depletion (**Fig EV5A** of the updated manuscript). 2) When overexpressed, BiP served as a repressor for both active IRE1 β/α (in absence of AGR2) and IRE1 α/α (upon ER stress induction) (**Fig EV5D** of the updated manuscript). Overall, these findings highlight that whilst AGR2 selectively represses the β isoform, BiP is more promiscuous and likely to partially contribute to IRE1 β regulation. This is in line with Bertolotti et al. 2000 Fig. 3E showing that BiP co-IPs with IRE1 β in a stress-dependent manner.*

2. Along the same lines, it could be useful to overexpress a more canonical PDI (e.g., PDIA1 or PDIA6 both of which have been implicated in regulating IRE1alpha) in the reporter cells to demonstrate the specificity of IRE1alpha/beta repression for AGR2.

*This point has been addressed experimentally. Quantification of flow cytometry analysis of transient transfection of PDIA1 in IRE1 β/α ; AGR2⁺ or IRE1 α/α expressing dual UPR reporter cells is depicted in **Fig 2D** of the updated manuscript. Whilst AGR2 expression repressed the deregulated IRE1 β/α , PDIA1 did not.*

3. Inclusion of some immunopurifications of the IRE1alpha/beta-AGR2 interactions in cells, especially as it relates to the mucin experiments, would help support the model described within the manuscript. Notably, one would predict that overexpression of mucin should deplete AGR2 from IRE1alpha/beta, which could be reflected by reduced interactions observed by IP. These experiments would provide additional evidence to support the idea that IRE1beta-AGR2 interaction is a specific stress sensor of mucin folding load within the ER.

A circuit involving endogenous AGR2 and MUC2 has been implicated in regulating IRE1b (in the accompanying manuscript by Cloots et al.) and has been reconstructed here by co-expression of MUC2 and AGR2 using the fluorescent reporters in cells. The reviewers suggested to complement these findings with co-immunoprecipitation (IP) experiments to test the prediction that abundance of an IRE1 β -AGR2 complex would correlate negatively with the mucin load in the ER. We accept the logic behind this suggestion but note that we are unable to detect an IRE1 β/α -AGR2 complex in cells expressing IRE1 β/α from the endogenous ERN1 locus. This likely reflects the low expression levels of the IRE1 β/α protein compounded by high off rates of the complex. This state of affairs all but preclude performing the experiment suggested by the reviewers in a system with a relevant concentration regime of IRE1 β/α .

Reviewer 2:

1) While overall, the study is convincing, the fact that the chimeric construct (IRE1beta luminal domain and IRE1alpha cytosolic region) is constitutively active, as reported in Figure 1, could also be caused by conformational perturbation of fusing two proteins and/or from two different species mouse and chinese hamster. The study of a cell line with the entire alpha locus replaced with the entire locus IRE1beta locus could provide excellent additional controls.

In the accompanying manuscript Cloots et al. shows that the intact IRE1 β , when expressed at even low levels in a heterologous system is constitutively active and subject to AGR2 repression (Fig 3A, therein), features mirrored by the IRE1 β/α chimera used here. Thus, we deem it unlikely that a corruption arising in the chimera gives rise to misleading experimental observations. Therefore, whilst we recognise the theoretical utility of the reviewer's suggestion, we note that creating such a cell line (via homologous recombination) is expected to take many months and therefore exceeds the time frame normally allowed for review. However, we have addressed the reviewer's suggestion editorially.

2) The study relies on a model line recapitulating proteins expressed in goblet cells. However, using a cellular model expressing IRE1beta and AGR2 could strengthen the physiological relevance of the study. In that model, it would be possible to delete AGR2 and confirm that no specific IRE1beta ligands are expressed under physiological expression.

This point was addressed in the accompanying manuscript Cloots et al Figure 3F-I.

3) I could not find the nature of the cells used for the transfections-IP in Figure 3. Are those CHO cells deficient for endogenous IRE1 alpha? Overexpression of chaperones has the tendency to generate artifacts due to the plasticity of the folds that can be expressed upon overexpression. If technically feasible, I suggest the authors confirm these experiments using the stable lines described in Figure 1 and the AGR2 ko described in the appendix as controls. Moreover, the parallel comparison of both BiP and AGR2 binding to both IRE1 alpha and IRE1 beta would provide excellent controls that could support the possible scenario: IRE1 alpha is BiP controlled and beta AGR2 controlled.

*We share the reviewer's concern regarding interpretation of experiments in which chaperones are overexpressed. Therefore to study the role of BiP, we emphasised its inactivation: Introduction of SubA, a protease that inactivates BiP by cleaving its interdomain linker (Paton et al. 2006) strongly activated IRE1 α/α signalling and the PERK dependent CHOP::GFP reporter. In contrast, IRE1 β/α appeared less subordinate to SubA-mediated BiP depletion (**Fig EV5A** of the updated manuscript). The conclusion drawn from manipulation of endogenous BiP levels were also supported by the observation that when overexpressed, BiP served as a repressor for both active IRE1 β/α (in absence of AGR2) and IRE1 α/α (upon ER stress induction) (**Fig EV5D** of the updated manuscript). Overall, these findings highlight that whilst AGR2 selectively represses the β isoform, BiP is more promiscuous and likely to partially contribute to IRE1 β regulation. This is in line with Bertolotti et al. 2000 (Fig 3E, therein) showing that BiP co-IPs with IRE1 β in a stress-dependent manner.*

The concern voiced by the reviewer also applies to IRE1 β and AGR2. However, as noted in the updated text we were unable to recover a complex between AGR2 and the low-abundant IRE1 β/α chimera by co-IP. In the accompanying paper, Cloots and

colleagues document such a complex in colon tissue lysates (where expression levels of both partners are much higher, Fig 2E therein). The analysis of IRE1 β -AGR2 complex in our paper was designed solely to test the role of mixed disulfide in complex formation, a question that when answered in the negative is unlikely to reflect an overexpression artefact (**Fig 3C** of the updated manuscript).

Elsewhere we have indicated the nature of the cells (wild-type CHO cells) used for IPs in **Fig 3**.

4) The KEN-mediated differences observed in Figure 8 are difficult to interpret. These differences in XBP1s signaling could be due to slight structural imperfections caused by the chimera in the construct with the beta intracellular portion. Because the LD and transmembrane region of alpha is conserved in both cases, it is difficult to exclude that transmission of the signal is more effective when the alpha region LD is fused to the alpha KEN region than when the alpha region LD is fused to the beta KEN region. A possible additional experiment that would support the finding would be to repeat the experiment using the beta LD domain and monitoring constitutive activation, for example. If the beta/beta construct is less active than the beta/alpha construct, it could support the idea that the nature of the response is different.

These are valid points. However, the main conclusion we draw from the comparison of the IRE1 α/α and IRE1 α/β chimera is their rather similar RIDD and XBP1 splicing activity. Thus, whilst both IRE1 isoforms are affected by divergent regulatory machineries, we found only modest differences comparing IRE1 α KEN vs IRE1 β KEN downstream signalling in our experimental system. These observations suggest that the driver for diversification in IRE1 function mainly arose from a specialisation in regulatory functions rather than effector functions. These considerations, together with the very significant effort posed by creating more knock-in cell lines, have led us to address this comment editorially.

Reviewer 3:

In expanded view Fig. EV3 the authors present elution profiles of size exclusion chromatography for IRE1 β in the absence of any interacting proteins or in the presence of different AGR2 wild type and mutant proteins. Normally the elution of column chromatography is given in volume units not time. It is curious that in panels A and B the time scale is 7 to 13 minutes whereas in panels C-E it is 7 to 13 seconds. The latter seems to be a typographic error.

We thank the reviewer for this comment and have corrected this typographic error in the figures. The analytical SEC experiments have been performed consistently at a constant flow rate of 0.3ml/min and therefore elution volume can be derived from elution time.

Fig. 6E the authors present the increase in IRE1 β dissociation rates with increasing AGR2 concentrations and fit the data to a one site binding equation with the dissociation rate of IRE1 β n dimers in the absence of AGR2 as background. They give the K1/2 max in units of $\mu\text{M}\cdot\text{s}^{-1}$. This is not correct. The resulting K1/2 max will be in units of μM , because in the denominator of the equation for the one site binding reaction there is only $\text{KD} + [\text{Ligand}]$ both of which should have the same units (mol/l).

We thank the reviewer for noticing this and have corrected this in the figure.

References:

Amin-Wetzel, N., Neidhardt, L., Yan, Y., Mayer, M.P., and Ron, D. (2019). Unstructured regions in IRE1alpha specify BiP-mediated destabilisation of the luminal domain dimer and repression of the UPR. *eLife* 8, e50793.

Bertolotti, A., Zhang, Y., Hendershot, L., Harding, H., and Ron, D. (2000). Dynamic interaction of BiP and the ER stress transducers in the unfolded protein response. *Nat Cell Biol* 2, 326-332.

Paton, A.W., Beddoe, T., Thorpe, C.M., Whisstock, J.C., Wilce, M.C., Rossjohn, J., Talbot, U.M., and Paton, J.C. (2006). AB5 subtilase cytotoxin inactivates the endoplasmic reticulum chaperone BiP. *Nature* 443, 548-552.

Yu, J., Li, T., Liu, Y., Wang, X., Zhang, J., Wang, X., Shi, G., Lou, J., Wang, L., Wang, C.C., *et al.* (2020). Phosphorylation switches protein disulfide isomerase activity to maintain proteostasis and attenuate ER stress. *EMBO J* 39, e103841.

Dear David and Lisa,

We have now received three reports on the revised version of your manuscript, which I have copied below. As you will see, you have addressed all the original concerns satisfactorily. Before I can finally accept the manuscript though, there are some remaining editorial points which need to be addressed. In this regard would you please:

- acknowledge funding for MRC Programme code (MC_U105184326) in our online submission system,
- rename the 'Conflict of interests' statement the 'disclosure and competing interest statement',
- remove the author credit section from the manuscript,
- consolidate all appendix figures to a single Appendix PDF, including a table of contents with page numbers,
- complete the Source Data checklist,
- include a 'Data Availability Section' that contains the following statement: 'This study includes no data deposited in external repositories.'
- define the annotated p values ***/** in the legend of figure 7d, and
- define 'n' and error bars in the legend of figure EV4e.

We include a synopsis of the paper (see <http://emboj.embopress.org/>). Please provide me with a two-sentence general summary statement and 3-5 bullet points that capture the key findings of the paper.

We also need a summary figure for the synopsis. The size should be 550 wide by [200-400] high (pixels). You can also use something from the figures if that is easier.

EMBO Press is an editorially independent publishing platform for the development of EMBO scientific publications.

Best wishes,

William

William Teale, PhD
Editor
The EMBO Journal
w.teale@embojournal.org

We realize that it is difficult to revise to a specific deadline. In the interest of protecting the conceptual advance provided by the work, we recommend a revision within 3 months (1st Feb 2024). Please discuss the revision progress ahead of this time with the editor if you require more time to complete the revisions. Use the link below to submit your revision:

Referee #1:

The authors have addressed all of my concerns from the previous submission. The manuscript is ready for publication in my opinion.

Referee #2:

The authors addressed most of the comments raised by the reviewers. I agree with the authors that the accompanying manuscript helps clarify aspects of the cellular role of AGR2 in regulating IRE1 β . Overall, this is a significant discovery supported by two complementary studies.

Referee #3:

The so-called unfolded protein response (UPR) is a homeostatic response to misfolded proteins in the endoplasmic reticulum and of eminent importance in health and disease. The elucidation of the molecular mechanism of the regulation of the UPR is therefore of great general interest across many fields of research including medical sciences. The study by Neidhardt et al. elucidated the molecular mechanism of Ire1 β a key regulator of the UPR in mucus producing cells using elegant and rigorous cell biological and biochemical experiments. The amount of data presented in this study is overwhelming and the quality convincing. In my opinion the authors have addressed all reviewers' comments adequately. Repeating experiments that are presented in the accompanying manuscript by Clout et al. seems unnecessary to me. This study constitutes a major advance in the field.

All editorial and formatting issues were resolved by the authors.

Dear David,

I am pleased to inform you that your manuscript has been accepted for publication in the EMBO Journal.

Thanks for trusting us to handle this really insightful study; and congratulations!

Best wishes,

William

William Teale, PhD
Editor
The EMBO Journal
w.teale@embojournal.org
